**Relevance**. Our work on online sequential decision-making with unknown delays is highly relevant to the topic of "ML for personalized search and recommendations" in the "User Modeling and Recommendation" track at WWW24 conference. The advent of large-scale applications, including but not limited to portfolio selection, online recommendation system, proactive resource allocation, online web ranking, and online shortest path planning, has generated substantial interest in the field of online sequential decision-making and convex optimization. In personalized search and recommendations, the goal is to provide users with tailored and relevant content based on their preferences and behavior. However, real-world systems often face challenges in dealing with dynamic and uncertain environments, such as unknown delays in feedback. Our work addresses this challenge by focusing on online sequential decision-making, where decisions need to be made in a sequential manner with limited information and unknown delays in the feedback loop. Based on the types of feedback information, we have designed three specific algorithms to handle function information, gradient information, and value information, respectively. These algorithms have been evaluated using universal norms and applied to various examples. The decision-making process of our online algorithms is guided by an approximate minimizer rather than an exact minimizer of the optimization problem. This approach is commonly employed in iterative optimization problems, as attaining infinite precision solutions for such problems is often unattainable. By developing algorithms and methodologies specifically designed for handling unknown delays, our research contributes to improving the effectiveness and efficiency of personalized search and recommendations. By considering the impact of unknown delays, our work enables more accurate and timely decision-making, which can lead to improved user satisfaction and engagement. It offers insights into how to adapt and optimize recommendation systems in real-time, taking into account delays that exist in online environments.

# Online Sequential Decision-Making with Unknown Delays

Anonymous Author(s)

## ABSTRACT

In the field of online sequential decision-making, we address the problem with delays utilizing the framework of online convex optimization (OCO), where the feedback of a decision can arrive with an unknown delay. Unlike previous research that is limited to Euclidean norm and gradient information, we propose three families of delayed algorithms based on approximate solutions to handle different types of received feedback. Our proposed algorithms are versatile and applicable to universal norms. Specifically, we introduce a family of Follow the Delayed Regularized Leader algorithms for feedback with full information on the loss function, a family of Delayed Mirror Descent algorithms for feedback with gradient information on the loss function and a family of Simplified Delayed Mirror Descent algorithms for feedback with the value information of the loss function's gradients at corresponding decision points. For each type of algorithm, we provide corresponding regret bounds under cases of general convexity and relative strong convexity, respectively. We also demonstrate the efficiency of each algorithm under different norms through concrete examples. Furthermore, our theoretical results are consistent with the current best bounds when degenerated to standard settings.

## CCS CONCEPTS

• **Computing methodologies → Online learning settings**.

## KEYWORDS

Sequential decision-making, Online convex optimization, Unknown delays, Approximate solution

**ACM Reference Format:**

Anonymous Author(s). 2023. Online Sequential Decision-Making with Unknown Delays. In *Proceedings of (International World Wide Web Conference)*. ACM, New York, NY, USA, 25 pages. https://doi.org/10.1145/nnnnnnn.nnnnnnn

## 1 INTRODUCTION

In modern society, a plethora of dynamic data has been generated in multiple domains, including Internet records, financial markets, consumer behaviors, and more. These data are revealed in a sequential manner and require rapid comprehension and updating. Correspondingly, online learning systems [3, 7] need real-time decision-making due to the continuous influx of new observational data. The advent of large-scale applications, including but not limited to portfolio selection [13], online recommendation system [25], proactive resource allocation [12], online web ranking [17], and online shortest path planning [18], has generated substantial interest in the field of online learning. Over the past few years, the integration of convex optimization techniques [6, 8] has brought about a transformation in the design of online learning algorithms, resulting in more efficient solutions and reliable theoretical analysis. Therefore, online convex optimization (OCO) [29, 30] has emerged as a powerful framework for modeling the problem of sequential decision-making. For instance, in the context of portfolio selection,

the strategy of an online investment aims to maximize returns by allocating the current wealth, without relying on any preconceived statistical assumptions. OCO is performed in a sequence of consecutive iterations. At each iteration $t$, the agent (investor) selects a decision $\boldsymbol{x}_t$ from a closed convex set (optional investment strategies) $\mathcal{X} \subseteq \mathbb{R}^n$. After submitting the decision, the agent receives information from the adversary regarding the loss function (market behavior) $f_t : \mathcal{X} \to \mathbb{R}$ and suffers an instantaneous loss (investment return) $f_t(\boldsymbol{x}_t)$. Due to the difficulty of attempting to maximize absolute wealth in an adversarial market, our algorithm can only maximize our wealth by comparing it to the returns achieved by a relatively advanced investment strategy and optimizing accordingly. The goal of the agent is to choose a sequence of decisions $\boldsymbol{x}_{[T]} = (\boldsymbol{x}_1, \ldots, \boldsymbol{x}_T)$ that minimizes the *regret*:

$$\text{Reg}_T := \sum_{t=1}^{T} f_t(\boldsymbol{x}_t) - \sum_{t=1}^{T} f_t(\boldsymbol{x}^*),$$

where $T$ is the time horizon, and $\boldsymbol{x}^* = \arg\min_{\boldsymbol{x} \in \mathcal{X}} \sum_{t=1}^{T} f_t(\boldsymbol{x})$ is the optimal decision in hindsight. In other words, regret is the gap between the cumulative loss of the agent and that incurred by a given sequence of comparators.

The standard framework of OCO assumes that the agent has immediate access to the information of the loss function $f_t$ before making the subsequent decision at iteration $t + 1$. In many practical scenarios, there exists a temporal gap between the decision-making process of the agent and the reception of the corresponding feedback. As an example of portfolio scenarios, it is typical to encounter unknown delays between making an investment and receiving returns, and the return cycles for different investments could also be asynchronous. Two variants of standard Online Gradient Descent (OGD) algorithm [14, 19], namely Delayed Online Gradient Descent (DOGD) algorithm [28] and Delayed Online Gradient Descent for Strongly Convex functions (DOGD-SC) [31], have been specifically designed to address the challenge of unknown delays in general convex and strongly convex functions, respectively.

However, these existing algorithms [28, 31] that solely rely on gradient feedback and Euclidean norm are not well-equipped to handle more universal scenarios competently. Firstly, in the online setting, adversaries who provide feedback to the agent typically withhold prior disclosure of the information type. Consequently, these gradient-based algorithms become ineffective once the feedback provided is not in the form of gradients.

Secondly, gradient descent-based algorithms may face challenges when computing projections efficiently for certain objective functions and constraint domain sets. As an example, let's consider the constant-rebalanced portfolio (CRP) consisting of $n$ stocks, the portfolio decision is represented by a probability distribution over the set of $n$ stocks. Computing the Euclidean projection onto the probability simplex could be computationally expensive.

Last but not least, theoretical analysis typically involves decomposing regret into a normal term caused by optimization steps and a delayed term caused by delayed feedback. The delayed term we

have established encompasses the dual norm of gradients, which actually live in the dual space, that is a distinct space from the primal space of decisions. The reason why these gradient-based delayed algorithms work effectively is that, in the specific case, the dual space coincides with the primal space. But, it is a very particular case that arises when the Euclidean norm is utilized. Instead, in universal cases, decisions and gradients exist in separate spaces.

Given the limitations of existing algorithms, in this paper, we propose three families of approximate algorithms based on types of the received feedback information. Particularly, in the universal space, we have specifically designed diverse regularization functions for different norms. These regularization functions quantify the divergence between variables in the primal and dual spaces and enable the mapping between these two spaces. Furthermore, we conduct an approximate solution in the optimization step, wherein the decision-making process in each iteration is guided by an approximate minimizer rather than an exact minimizer of the optimization problem. This approach is commonly utilized in iterative optimization problems because it is often impractical to achieve solutions with infinite precision.

Our contributions are summarized as follows:

- Firstly, we propose three types of online algorithms to handle delayed feedback. Our theoretical analysis differs from gradient descent-based algorithms as it involves addressing universal norms and approximate solutions, which poses a challenge.
- For the full information feedback of loss functions, we propose a family of follow the delayed regularized leader algorithms to handle the general convex functions and relative strongly convex functions, respectively. Notably, we replace the conventional concept of strong convexity with the more general notion of relative strong convexity.
- For the gradient information feedback, we introduce a family of delayed mirror descent algorithms. Moreover, we showcase the versatility and superiority of our proposed algorithms by applying them to examples with various norms.
- When feedback is limited to the value information of loss functions' gradients, we introduce a family of simplified delayed mirror descent algorithms. we demonstrate that despite the reduced amount of information obtained, it can achieve regret bounds comparable to those obtained with full information or gradient information.

*Organization.* We mention the existing work related to our paper in Section 2. In Section 3, we introduce the formal definitions, assumptions and examples. In Section 4, we present a family of follow the delayed regularized leader algorithms based on full information feedback for both general and relative strongly convex cases, and also provide examples to illustrate their performance. In Section 5, we utilize the delayed gradient information to design a family of delayed mirror descent algorithms. In Section 6, we develop a family of simplified delayed mirror descent algorithms that can handle situations where feedback is reduced to the value information of loss functions' gradients. We end off the paper with the conclusion and future work in Section 7. Additionally, we provide a detailed theoretical analysis in the appendix.

## 2 RELATED WORK

### 2.1 The Standard OCO

When $d_t = 1$ for each $t \in [T]$, our problem is degenerated to the standard framework of OCO [9, 30]. In the context of OCO, the type of feedback received by the agent is a crucial aspect in the development of online learning algorithms. When the feedback consists of the full information of loss functions, a natural approach is to select the decision that optimizes the loss history for all previous iterations, i.e., $x_{t+1} = \arg\min_{x \in \mathcal{X}} \sum_{\tau=1}^{t} f_\tau(x)$, commonly known as the Follow the Leader (FTL) algorithm [1, 30]. Furthermore, to ensure stability and prevent oscillations between decisions, the Follow the Regularized Leader (FTRL) algorithm [23, 27] selected the decision that minimizes the sum of previous losses and an additional regularization term $\psi$ with a learning rate $\eta_t$ as follows:

$$x_{t+1} = \arg\min_{x \in \mathcal{X}} \left\{ \sum_{\tau=1}^{t} f_\tau(x) + \frac{1}{\eta_t}\psi(x) \right\}, \quad (1)$$

When the revealed feedback is in the form of gradient information of the loss function, a helpful way to understand the algorithm's performance is to perceive it as minimizing a local estimate of the original loss function. By the definition of convexity, a linear bound for the function $f_t$ around $x_t$ could be constructed by

$$\forall x \in \mathcal{X}, \tilde{f}_t(x) := f_t(x_t) + \langle \nabla f_t(x_t), x - x_t \rangle.$$

Unfortunately, the direct minimization of a linear function may not lead to an effective online algorithm since the minimum of a linear function can approach negative infinity over unbounded domains. This issue could be addressed by confining the minimization of the lower bound within a specific neighborhood of $x_t$. Online Mirror Descent (OMD) algorithm [5, 15] achieved this by utilizing the Bregman divergence, as illustrated below:

$$x_{t+1} = \arg\min_{x \in \mathcal{X}} \left\{ \langle \nabla f_t(x_t), x \rangle + \frac{1}{\eta_t} B_\psi(x; x_t) \right\}. \quad (2)$$

Both FTRL and OMD have been shown to obtain regret bounds of $O(\sqrt{T})$ and $O(\ln T)$ for general convex and strongly convex functions, respectively. Notably, when the regularization term is specified as $\psi = \frac{1}{2}\|\cdot\|_2^2$, OMD is equivalent to the well-studied projection-based OGD [4, 34], which employed the following update rule: $x_{t+1} = \Pi_{\mathcal{X}}(x_t - \eta_t \nabla f_t(x_t))$, where $\Pi_{\mathcal{X}}(\cdot)$ is the Euclidean projection onto the closed convex set $\mathcal{X}$.

Despite the extensive research on standard OCO algorithms, the practical issue of delayed feedback has not been adequately addressed.

### 2.2 The Delayed OCO

For the scenario where all delays are fixed and known, i.e., $d_t = \bar{d}$ for each $t \in [T]$, the seminal work of [32] addressed this issue by dividing the total $T$ iterations into $\bar{d}$ subsets and maintaining a base algorithm on each subset. By utilizing the standard OGD algorithm as the base, [32] achieved an $O(\sqrt{\bar{d}T})$ regret bound for general convex functions. [35] investigated the same delayed case and demonstrated that incorporating delayed gradients during each gradient descent step in the standard OGD algorithm also yields regret bounds of $O(\sqrt{\bar{d}T})$ and $O(\bar{d}\ln T)$ for convex losses and strongly convex functions, respectively.

Moreover, [22] extended the approach proposed in [32] to handle a more challenging scenario where delays are arbitrary but time-stamped and achieved a regret bound of $O(\sqrt{dT})$, where $d$ denotes the maximum delay. In a similar vein, [21] employed the one-point gradient estimator [16] to introduce a comparable approach for the bandit setting.

In the most general delayed case, where each feedback could be delayed arbitrarily and the time stamp of each feedback could also be unknown. [26] presented a data-dependent regret bound for the delayed setting, assuming the decision set is unbounded. The celebrated work [28] proposed an effective method named DOGD and obtained a regret bound of $O(\sqrt{D_T})$, where $D_T$ is the cumulative delays. DOGD, given by

$$x_{t+1} = \Pi_X \left[ x_t - \eta_t \sum_{k \in \mathcal{F}_t} \nabla f_k(x_k) \right],$$

needs to query the gradient at each iteration $t$ and update the decision with the sum of those gradients queried at the set of iterations $\mathcal{F}_t$. [24] proposed DBGD, which employed an $(n+1)$-point gradient estimator [2] to approximate gradients in the bandit setting. [10] expanded [28] to the decentralized optimization over a network. [31] adopted a time-varying learning rate connected with the total number of observable feedback to address the delayed optimization problem with strongly convex loss functions and obtained an $O(d \ln T)$ regret bound.

To date, most studies on delayed OCO have relied on the Euclidean norm to measure the distance between the decision points. However, it is unclear whether these algorithms can be extended to accommodate other types of norms. This paper gives an affirmative answer by proposing three families of FTDRL, DMD and SDMD algorithms.

Compared to the previous delayed gradient descent-based algorithms, our proposed algorithms exhibit advancements in the following aspects. Firstly, we introduce three targeted algorithms that address different types of feedback related to function information, gradient information, and value information, respectively. Secondly, our algorithms are grounded on universal norms, instead of a specific Euclidean norm, and demonstrate superiority across diverse examples. Thirdly, we employ an approximate solution approach for each optimization problem. Besides, we adopt a theoretical analysis method that is entirely different from previous works, specifically analyzing the regret bounds when the loss function is either general convex or more flexible relative strongly convex.

## 3 FORMAL NOTATIONS

First, to provide a more concrete definition of the delayed setting, we introduce the following notation. Let $d_t \in \mathbb{Z}^+$ denote a non-negative integral delay at iteration $t$. At the end of iteration $t+d_t-1$, the feedback queried at iteration $t$ is received and can be used in iteration $t + d_t$. In the standard setting, where there are no delays, $d_t = 1$ for all $t$. We denote the set of iterations that receive feedback at the end of iteration $t$ as $\mathcal{F}_t$. The maximum delay and the total delay (up to iteration $T$) are denoted by $d = \max_{t \in [T]} d_t$ and $D_T = \sum_{t=1}^{T} d_t$, respectively.

In this paper, we denote the $n$-dimensional real vector space equipped with an inner product $\langle \cdot, \cdot \rangle$ and a norm $\| \cdot \|$ by $\mathbb{R}^n$. The dual norm of $\| \cdot \|$ is defined as $\|x\|_\star := \sup_{y \in \mathbb{R}^n : \|y\| \leq 1} \langle x, y \rangle$ for each $x \in \mathbb{R}^n$. We use the notation $X \subseteq \mathbb{R}^n$ to denote a closed convex set and $\{f_t : X \to \mathbb{R}\}_{t \geq 1}$ to denote a sequence of convex loss functions. Moreover, let $\psi : \mathcal{D} \to \mathbb{R}$ be a convex function such that it is differentiable in its non-empty interior $\mathcal{D}^\circ := \text{int } \mathcal{D}$ and such that we have $X \subseteq \mathcal{D}^\circ$.

DEFINITION 1. *For a convex function $f : X \to \mathbb{R}$ and $x \in X$, a vector $\nabla f(x) \in \mathbb{R}^n$ is the gradient of $f$ at $x$, then $\nabla f(x)$ satisfies the inequality*

$$\forall y \in X, f(y) \geq f(x) + \langle \nabla f(x), y - x \rangle.$$

DEFINITION 2. *The Bregman divergence with respect to function $\psi$ is given by*

$$\forall x \in \mathcal{D}, y \in \mathcal{D}^\circ, B_\psi(x; y) = \psi(x) - \psi(y) - \langle \nabla \psi(y), x - y \rangle.$$

ASSUMPTION 1. *The primal norm of the decision is bounded by $R$ and the dual norm of the gradient is bounded by $G_\star$, i.e.,*

$$\forall x \in X, t \in [T], \|x\| \leq R, \|\nabla f_t(x)\|_\star \leq G_\star.$$

ASSUMPTION 2. *The regularization function $\psi$ has $G_\psi$-Lipschitz gradients on the set $X$, i.e.,*

$$\forall x, y \in \mathcal{D}, \|\nabla \psi(x) - \nabla \psi(y)\|_\star \leq G_\psi \|x - y\|.$$

For convenience, we make $G_\psi = \xi G_\star$.

ASSUMPTION 3. *The regularization function $\psi$ is $\sigma$-strongly convex over $\mathcal{D}$ with respect to a norm $\| \cdot \|$, i.e.,*

$$\forall x \in \mathcal{D}, y \in \mathcal{D}^\circ, B_\psi(x; y) \geq \frac{\sigma}{2} \|x - y\|^2.$$

ASSUMPTION 4. *(**Relative strong convexity**) If the loss function $f_t$ is $\gamma$-strongly convex over $X$ with respect to a convex and differentiable function $\psi$, then,*

$$\forall x, y \in X, t \in [T], f_t(x) - f_t(y) - \langle \nabla f_t(y), x - y \rangle \geq \gamma B_\psi(x; y).$$

A noteworthy instance of relative strong convexity occurs when we select $\psi(x) = \frac{1}{2} \|x\|_2^2$, and the classical strong convexity definition with respect to the Euclidean norm is recovered.

To showcase the effectiveness of our algorithms when applied to different norms, we provide the following examples, each of which corresponds to a distinct domain and regularizer. The choice of the regularizer is primarily determined by its strong convexity with respect to a particular norm.

EXAMPLE 1. *In the Euclidean space, we consider $X \in \mathbb{R}^n$ and $\psi(x) = \frac{1}{2} \|x\|_2^2$. Note that the dual norm of $\| \cdot \|_2$ is itself, and $B_\psi(x; y) = \frac{1}{2} \|x - y\|_2^2$. Moreover, $\psi(\cdot)$ is 1-strongly convex with respect to norm $\| \cdot \|_2$ over $X$. We assume that $\|\nabla f_t(x)\|_2 \leq G_2$ and $\|x\|_2 \leq R_2$ for any $x \in X, t \in [T]$.*

EXAMPLE 2. *In the probabilistic simplex, we consider $X = \{x \in \mathbb{R}_+^n : \|x\|_1 = 1\}$. and $\psi(x) = \sum_{\mu=1}^{n} x^{(\mu)} \ln x^{(\mu)} + \ln n$. The dual norm of $\| \cdot \|_1$ is $\| \cdot \|_\infty$. Note that $\psi(\cdot)$ is 1-strongly convex with $\| \cdot \|_1$. We assume that $\|\nabla f_t(x)\|_\infty \leq G_\infty$ for any $x \in X, t \in [T]$.*

---

**Algorithm 1** Follow the Delayed Regularized Leader for General Convexity

---

1: Input: both $\eta$ and $x_1$ depend on the choice of example
2: **for** $t = 1, \ldots, T$ **do**
3:     Query $f_t$ and receive feedback $\{f_k : k \in \mathcal{F}_t\}$
4:     **if** $|\mathcal{F}_t| > 0$ **then**
5:         Make an approximate solution $x_{t+1}$:

$$\sum_{\tau=1}^{t} \sum_{k \in \mathcal{F}_t} f_k(x_{t+1}) + \frac{\psi(x_{t+1})}{\eta} \le \sum_{\tau=1}^{t} \sum_{k \in \mathcal{F}_t} f_k(y_{t+1}^*) + \frac{\psi(y_{t+1}^*)}{\eta} + \rho_t,$$

$$\text{where } y_{t+1}^* = \arg\min_{x \in \mathcal{X}} \left\{ \sum_{\tau=1}^{t} \sum_{k \in \mathcal{F}_t} f_k(x) + \frac{1}{\eta} \psi(x) \right\}$$

6:     **else**
7:         $x_{t+1} = x_t$
8:     **end if**
9: **end for**

---

**Algorithm 2** Follow the Delayed Leader for Relative Strong Convexity

---

1: Input: make an arbitrary decision $x_1 \in \mathcal{X}$
2: **for** $t = 1, \ldots, T$ **do**
3:     Query $f_t$ and receive feedback $\{f_k : k \in \mathcal{F}_t\}$
4:     **if** $|\mathcal{F}_t| > 0$ **then**
5:         Make an approximate solution $x_{t+1}$:

$$\sum_{\tau=1}^{t} \sum_{k \in \mathcal{F}_t} f_k(x_{t+1}) \le \sum_{\tau=1}^{t} \sum_{k \in \mathcal{F}_t} f_k(y_{t+1}^*) + \rho_t,$$

$$\text{where } y_{t+1}^* = \arg\min_{x \in \mathcal{X}} \left\{ \sum_{\tau=1}^{t} \sum_{k \in \mathcal{F}_t} f_k(x) \right\}$$

6:     **else**
7:         $x_{t+1} = x_t$
8:     **end if**
9: **end for**

---

EXAMPLE 3. *In the case of p-norm, we consider $\psi(x) = \frac{1}{2}\|x\|_p^2$ over $\mathcal{X} \in \mathbb{R}^n$, where $\|x\|_p = \left( \sum_{\mu=1}^{n} |x^{(\mu)}|^p \right)^{\frac{1}{p}}$ and $1 < p \le 2$. Note that the dual norm of $\|\cdot\|_p$ is $\|\cdot\|_q$, where $\frac{1}{p} + \frac{1}{q} = 1$. $\psi(\cdot)$ is $(p-1)$-strongly convex with respect to the norm $\|\cdot\|_p$. We assume $\|x\|_p \le R_p$ and $\|\nabla f_t(x)\|_q \le G_q$ for any $x \in \mathcal{X}, t \in [T]$.*

# 4 FOLLOW THE DELAYED REGULARIZED LEADER

When an agent has access to the full information of loss functions at each iteration, the FTRL algorithm is a commonly used approach. Motivated by the standard FTRL algorithm presented in Eq. (1), which does not incorporate delays, we replace the standard history from beginning to iteration $t$ with the outdated history of revealed loss functions in the iteration set $\{\mathcal{F}_\tau : \tau \in [t]\}$.

## 4.1 Sublinear Regret for General Convexity

When the loss function is general convex, we propose a Follow the Delayed Regularized Leader for General Convexity (FTDRL-GC) algorithm. The detailed update procedures are summarized in

Algorithm 1. The optimization problem arising in our algorithm is only required to be solved approximately (up to an additive error $\rho_t$). This is commonly observed in iterative optimization problems, since in general they cannot be solved exactly. Moreover, approximating the optimization problem induces a sequence of errors that complicates the regret analysis of the algorithm.

By setting a constant learning rate, the regret bound is formally stated in the following theorem.

THEOREM 1. *Under Assumptions 1 and 3, let the maximum approximate error $\rho_t = \frac{\eta G_\star^2}{8\sigma}, \forall t \in [T]$, Algorithm 1 satisfies*

$$\text{Reg}_T \le \frac{\eta G_\star^2 (T + 4D_T)}{\sigma} + \frac{\psi(x^*) - \psi(x_1)}{\eta}.$$

PROOF. For convenience, we set $\Phi_{t,i_k}(x) = \sum_{\tau=1}^{t-1} \sum_{s \in \mathcal{F}_\tau} f_s(x) + \sum_{s \in \mathcal{F}_{t,k}} f_s(x) + f_k(x) + \frac{1}{\eta}\psi(x)$ and $\Phi_0(x) = \frac{1}{\eta}\psi(x)$. At each iteration $t$ in FTDRL-GC, for each $k \in \mathcal{F}_t$, we view the update process as $|\mathcal{F}_t|$ segments as

$$\Phi_{t,i_k}(x_{t,i_k+1}) \le \Phi_{t,i_k}(y_{t,i_k+1}^*) + \rho_{t,i_k}, y_{t,i_k+1}^* = \arg\min_{x \in \mathcal{X}} \Phi_{t,i_k}(x),$$

where $i_k = |\mathcal{F}_{t,k}|, \mathcal{F}_{t,k} = \{s \in \mathcal{F}_t : s < k\}$. Additionally, we make $x_{t,0} = x_t, x_{t+1} = x_{t,|\mathcal{F}_t|}$, and $\rho_{t,0} = \rho_t, \rho_{t+1} = \rho_{t,|\mathcal{F}_t|}$. The total regret bound can be divided into two parts.

$$\text{Reg}_T = \sum_{t=1}^{T} [f_t(x_t) - f_t(x^*)] = \sum_{t=1}^{T} \sum_{k \in \mathcal{F}_t} [f_k(x_k) - f_k(x^*)]$$

$$= \underbrace{\sum_{t=1}^{T} \sum_{k \in \mathcal{F}_t} [f_k(x_{t,i_k}) - f_k(x^*)]}_{\text{normal term}} + \underbrace{\sum_{t=1}^{T} \sum_{k \in \mathcal{F}_t} [f_k(x_k) - f_k(x_{t,i_k})]}_{\text{delayed term}}.$$

$$(3)$$

For the normal term of Eq. (3), we have the following lemma.

LEMMA 1. *Under Assumptions 1 and 3, the normal term of Eq. (3) is bounded by*

$$\sum_{t=1}^{T} \sum_{k \in \mathcal{F}_t} [f_k(x_{t,i_k}) - f_k(x^*)] \le \frac{\eta}{\sigma} T G_\star^2 + \frac{1}{\eta} [\psi(x^*) - \psi(x_1)]. \quad (4)$$

Next, we discuss the delayed term of Eq. (3), we have

$$\sum_{t=1}^{T} \sum_{k \in \mathcal{F}_t} [f_k(x_k) - f_k(x_{t,i_k})] \le \sum_{t=1}^{T} \sum_{k \in \mathcal{F}_t} G_\star \|x_k - x_{t,i_k}\|$$

$$\le \sum_{t=1}^{T} \sum_{k \in \mathcal{F}_t} G_\star \left( \sum_{\tau=k}^{t-1} \sum_{s \in \mathcal{F}_\tau} \|x_{\tau,i_s+1} - x_{\tau,i_s}\| + \sum_{s \in \mathcal{F}_{t,k}} \|x_{t,i_s+1} - x_{t,i_s}\| \right).$$

$$(5)$$

The crucial factor impacting the bound of the delayed term is the gap between $x_{\tau,i_s}$ and $x_{\tau,i_s+1}$.

LEMMA 2. *Under Assumptions 1 and 3, for each $t \in [T], k \in \mathcal{F}_t$, our FTDRL-GC algorithm ensures that*

$$\|x_{t,i_k+1} - x_{t,i_k}\| \le \frac{2\eta G_\star}{\sigma}. \quad (6)$$

Substituting Eq. (6) into Eq. (5) gives

$$\sum_{t=1}^{T} \sum_{k \in \mathcal{F}_t} [f_k(\boldsymbol{x}_k) - f_k(\boldsymbol{x}_{t,i_k})] \leq \frac{4\eta G_\star^2 D_T}{\sigma}. \tag{7}$$

The last inequality is derived from the following lemma.

LEMMA 3 ([28]). *The summation terms of the delay satisfy*

$$\sum_{t=1}^{T} \sum_{k \in \mathcal{F}_t} \left( \sum_{\tau=k}^{t-1} |\mathcal{F}_\tau| + |\mathcal{F}_{t,k}| \right) \leq 2D_T.$$

Combining with Eq. (4) and Eq. (7) gives the result of Theorem 1. □

Intuitively, the regret bound in Theorem 1 is likely to be influenced by the specific characteristics of different examples (i.e., the choice of learning rate $\eta$ and initial decision $\boldsymbol{x}_1$ based on the nature of specific regularization function). To illustrate this point, we give the following corollary.

COROLLARY 1. *Applying FTDRL-GC to Example 1 with $\boldsymbol{x}_1 = \boldsymbol{0} \in \mathcal{X}$ and $\eta = \frac{R_2}{G_2} \sqrt{\frac{1}{2T+8D_T}}$ yields $\mathrm{Reg}_T \leq G_2 R_2 \sqrt{2T + 8D_T}$.*

*Applying FTDRL-GC to Example 2 with $\boldsymbol{x}_1 = [\frac{1}{n}, \dots, \frac{1}{n}] \in \mathbb{R}_+^n$ and $\eta = \frac{1}{G_\infty} \sqrt{\frac{\ln n}{T+4D_T}}$ yields $\mathrm{Reg}_T \leq 2G_\infty \sqrt{(T + 4D_T) \ln n}$.*

*Applying FTDRL-GC to Example 3 with $\boldsymbol{x}_1 = \boldsymbol{0} \in \mathcal{X}$ and $\eta = \frac{R_p}{G_q} \sqrt{\frac{p-1}{2T+8D_T}}$ yields $\mathrm{Reg}_T \leq R_p G_q \sqrt{\frac{2T+8D_T}{p-1}}$.*

We can naturally employ the doubling trick [11] to address the scenarios where the values of $D_T$ and $T$ are unknown. This technique also guarantees that the maximum permissible errors in the approximate solution continuously decrease from their initially larger values.

*Remark.* When utilizing the Euclidean norm with regularizer $\psi(\boldsymbol{x}) = \frac{1}{2}\|\boldsymbol{x}\|^2$, the results achieved are comparable to those obtained in the DOGD algorithm [28]. However, in the case of the probability simplex, we can achieve a regret bound of $O(G_\infty \sqrt{D_T \ln n})$. In contrast to the DODG algorithm applied in this scenario, which yielded a regret of $O(G_2 \sqrt{D_T})$, for the worst-case scenario (i.e., $G_2 = \sqrt{n} G_\infty$). We transforming the dimension dependency from $\sqrt{n}$ to $\sqrt{\ln n}$. Additionally, our regrets match the regime of standard FTRL [23] when considering $d_t = 1, \forall t \in [T]$.

## 4.2 Logarithmic Regret for Relative Strong Convexity

[20, 33] have shown that the non-delayed FTL algorithm without any regularization function achieves logarithmic regret in the case of relative strong convexity. Inspired by these works, we propose a Follow the Delayed Leader for Relative Strong Convexity (FTDL-RSC) algorithm in the presence of delays and formally outline the update procedures in Algorithm 2. At each iteration $t$, we directly minimize the outdated history over the iteration set $\{\mathcal{F}_\tau : \tau \in [t]\}$. Compared with previous delayed algorithm [31], our FTDL-RSC algorithm possesses several advantages in three aspects. Firstly, we are capable of computing an approximate solution for each decision. Secondly, we can handle universal norms. Thirdly, we do not need prior knowledge of the modules of relative strong convexity.

We establish the following theorem and corollary regarding the regret bound of Algorithm 2.

THEOREM 2. *Under Assumptions 1, 3 and 4, let the maximum approximate error $\rho_t = \frac{|\mathcal{F}_t|^2 G_\star^2}{8 \sum_{\tau=1}^{t} |\mathcal{F}_\tau| \gamma \sigma}, \forall t \in [T]$, Algorithm 2 satisfies*

$$\mathrm{Reg}_T \leq \frac{3dG_\star^2}{\sigma \gamma}(1 + \ln T).$$

PROOF. In the case of relative strong convexity, the regret of FTDL-RSC is divided by

$$\mathrm{Reg}_T = \underbrace{\sum_{t=1}^{T} \sum_{k \in \mathcal{F}_t} [f_k(\boldsymbol{x}_t) - f_k(\boldsymbol{x}^*)]}_{\text{normal term}} + \underbrace{\sum_{t=1}^{T} \sum_{k \in \mathcal{F}_t} [f_k(\boldsymbol{x}_k) - f_k(\boldsymbol{x}_t)]}_{\text{delayed term}}. \tag{8}$$

For the normal term of Eq. (8), we have the following lemma.

LEMMA 4. *Under Assumptions 1,3 and 4, the normal term of Eq. (8) is bounded by*

$$\sum_{t=1}^{T} \sum_{k \in \mathcal{F}_t} [f_k(\boldsymbol{x}_t) - f_k(\boldsymbol{x}^*)] \leq \frac{dG_\star^2(1 + \ln T)}{\sigma \gamma}. \tag{9}$$

Next, we analyze the delayed term of Eq. (8).

$$\sum_{t=1}^{T} \sum_{k \in \mathcal{F}_t} [f_k(\boldsymbol{x}_k) - f_k(\boldsymbol{x}_t)] \leq \sum_{t=1}^{T} \sum_{k \in \mathcal{F}_t} \langle \nabla f_k(\boldsymbol{x}_k), \boldsymbol{x}_k - \boldsymbol{x}_t \rangle$$

$$\leq \sum_{t=1}^{T} \sum_{k \in \mathcal{F}_t} G_\star \left( \sum_{\tau=k}^{t-1} \|\boldsymbol{x}_{\tau+1} - \boldsymbol{x}_\tau\| \right) \leq dG_\star \sum_{t=1}^{T} \|\boldsymbol{x}_{t+1} - \boldsymbol{x}_t\|. \tag{10}$$

The last inequality holds because $d$ is the maximum delay.

Here, we discuss the difference between $\boldsymbol{x}_t$ and $\boldsymbol{x}_{t+1}$.

LEMMA 5. *Under Assumptions 1,3 and 4, for each $t \in [T]$, our FTDL-RSC algorithm ensures that*

$$\|\boldsymbol{x}_{t+1} - \boldsymbol{x}_t\| \leq \frac{3}{2} \frac{|\mathcal{F}_t| G_\star}{\sum_{\tau=1}^{t} |\mathcal{F}_\tau| \sigma \gamma} + \frac{1}{2} \frac{|\mathcal{F}_{t-1}| G_\star}{\sum_{\tau=1}^{t-1} |\mathcal{F}_\tau| \sigma \gamma}. \tag{11}$$

Substituting Eq. (11) into Eq. (10) gives the delayed term.

$$\sum_{t=1}^{T} \sum_{k \in \mathcal{F}_t} [f_k(\boldsymbol{x}_k) - f_k(\boldsymbol{x}_t)] \leq \frac{2dG_\star^2(1 + \ln T)}{\sigma \gamma}. \tag{12}$$

Combining with Eq. (9) and Eq. (12), we obtain Theorem 2. □

COROLLARY 2. *Applying FTDL-RSC to Example 1 gives $\mathrm{Reg}_T \leq \frac{3dG_2^2}{\gamma}(1 + \ln T)$.*

*Applying FTDL-RSC to Example 2 gives $\mathrm{Reg}_T \leq \frac{3dG_\infty^2}{\gamma}(1 + \ln T)$.*

*Applying FTDL-RSC to Example 3 gives $\mathrm{Reg}_T \leq \frac{3dG_q^2}{\gamma(p-1)}(1 + \ln T)$.*

*Remark.* Above reveals that when applying the Euclidean norm with the regularization function $\psi(\boldsymbol{x}) = \frac{1}{2}\|\boldsymbol{x}\|^2$, our FTDL-RSC achieves regret bounds that match those of the DOGD-SC algorithm [31]. However, in comparison to directly applying DOGD-SC to the probability simplex scenario, our approach, which utilizes an entropic regularization function, leads to an improved dependency of the regret bound from $G_2$ to $G_\infty$. Furthermore, assuming

---

**Algorithm 3** Delayed Mirror Descent for General Convexity

1: Input: both $\eta$ and $x_1$ depend on the choice of example
2: **for** $t = 1, \ldots, T$ **do**
3:     Query $\nabla f_t$ and receive feedback $\{\nabla f_k : k \in \mathcal{F}_t\}$
4:     **if** $|\mathcal{F}_t| > 0$ **then**
5:       $x_{t,0} = x_t$
6:       **for** $k \in \mathcal{F}_t$ **do**
7:         Make an approximate solution $x_{t,i_k+1}$:

$$\left\langle \nabla f_k(x_{t,i_k}), x_{t,i_k+1} \right\rangle + \frac{1}{\eta} B_\psi(x_{t,i_k+1}; x_{t,i_k})$$

$$\leq \left\langle \nabla f_k(x_{t,i_k}), y^*_{t,i_k+1} \right\rangle + \frac{1}{\eta} B_\psi(y^*_{t,i_k+1}; x_{t,i_k}) + \rho_{t,i_k},$$

$$\text{where } y^*_{t,i_k+1} = \arg\min_{x \in \mathcal{X}} \left\{ \left\langle \nabla f_k(x_{t,i_k}), x \right\rangle + \frac{1}{\eta} B_\psi(x; x_{t,i_k}) \right\}$$

8:       **end for**
9:       $x_{t+1} = x_{t,|\mathcal{F}_t|}$
10:     **else**
11:       $x_{t+1} = x_t$
12:     **end if**
13: **end for**

---

$d_t = 1, \forall t \in [T]$, our findings align with the regret bound of the non-delayed FTL algorithm [33] for relative strong convexity.

## 5 DELAYED MIRROR DESCENT

When the feedback available is the gradient information of the loss functions, we develop a family of delayed mirror descent for the general convexity and relative strong convexity, respectively. Motivated by the standard OMD formulation given in Eq. (2), we replace the gradient $\{\nabla f_t\}$ with the gradient set $\{\nabla f_k : k \in \mathcal{F}_t\}$ at each iteration $t$ for updates.

### 5.1 Sublinear Regret for General Convexity

Here we propose a Delayed Mirror Descent for General Convexity (DMD-GC) algorithm. The detailed procedure is summarized in Algorithm 3. The update process is divided into $|\mathcal{F}_t|$ segments and a fixed learning rate $\eta$ is utilized. The index $i_k = |\mathcal{F}_{t,k}|$ denotes the position of iteration $k$ in set $\mathcal{F}_t$ where $\mathcal{F}_{t,k} = \{s \in \mathcal{F}_t : s < k\}$. Similarly, each decision is made based on an approximate solution.

We establish the following theorem regarding the regret bound.

THEOREM 3. *Under Assumptions 1, 2 and 3, let the maximum approximate error $\rho_{t,i_k} = \frac{\eta^3}{2\sigma}, \forall k \in \mathcal{F}_t, t \in [T]$, Algorithm 3 satisfies*

$$\text{Reg}_T \leq \frac{\eta(G_\star^2 T + 8\xi RG_\star T + 2\eta G_\star T + 4G_\star^2 D_T + 8\eta G_\star D_T)}{2\sigma}$$

$$+ \frac{2\eta^2 \xi G_\star^2 D_T}{\sigma^2} + \frac{B_\psi(x^*; x_1)}{\eta}.$$

PROOF. The total regret bound is divided by

$$\text{Reg}_T = \underbrace{\sum_{t=1}^{T} \sum_{k \in \mathcal{F}_t} [f_k(x_{t,i_k}) - f_k(x^*)]}_{\text{normal term}} + \underbrace{\sum_{t=1}^{T} \sum_{k \in \mathcal{F}_t} [f_k(x_k) - f_k(x_{t,i_k})]}_{\text{delayed term}}.$$

$$(13)$$

---

**Algorithm 4** Delayed Mirror Descent for Relative Strong Convexity

1: Input: $\eta_t = \frac{1}{\gamma \sum_{\tau=1}^{t} |\mathcal{F}_\tau|}$, make an arbitrary decision $x_1 \in \mathcal{X}$
2: **for** $t = 1, \ldots, T$ **do**
3:     Query $\nabla f_t$ and receive feedback $\{\nabla f_k : k \in \mathcal{F}_t\}$
4:     **if** $|\mathcal{F}_t| > 0$ **then**
5:       Make an approximate solution $x_{t+1}$:

$$\sum_{k \in \mathcal{F}_t} \left\langle \nabla f_k(x_t), x_{t+1} \right\rangle + \frac{1}{\eta_t} B_\psi(x_{t+1}; x_t)$$

$$\leq \sum_{k \in \mathcal{F}_t} \left\langle \nabla f_k(x_t), y^*_{t+1} \right\rangle + \frac{1}{\eta_t} B_\psi(y^*_{t+1}; x_t) + \rho_t,$$

$$\text{where } y^*_{t+1} = \arg\min_{x \in \mathcal{X}} \left\{ \left\langle \sum_{k \in \mathcal{F}_t} \nabla f_k(x_t), x \right\rangle + \frac{1}{\eta_t} B_\psi(x; x_t) \right\}$$

6:     **else**
7:       $x_{t+1} = x_t$
8:     **end if**
9: **end for**

---

For the normal term of Eq. (13), we have the following lemma.

LEMMA 6. *The normal term of Eq. (13) is bounded by*

$$\sum_{t=1}^{T} \sum_{k \in \mathcal{F}_t} [f_k(x_{t,i_k}) - f_k(x^*)] \leq \frac{\eta T(G_\star^2 + 8\xi RG_\star + 2\eta G_\star)}{2\sigma} + \frac{B_\psi(x^*; x_1)}{\eta}.$$

$$(14)$$

Next, we discuss the delayed term of Eq. (13).

$$\sum_{t=1}^{T} \sum_{k \in \mathcal{F}_t} [f_k(x_k) - f_k(x_{t,i_k})] \leq \sum_{t=1}^{T} \sum_{k \in \mathcal{F}_t} \left\langle \nabla f_k(x_k), x_k - x_{t,i_k} \right\rangle$$

$$\leq \sum_{t=1}^{T} \sum_{k \in \mathcal{F}_t} G_\star \left( \sum_{\tau=k}^{t-1} \sum_{s \in \mathcal{F}_\tau} \|x_{\tau,i_s+1} - x_{\tau,i_s}\| + \sum_{s \in \mathcal{F}_{t,k}} \|x_{t,i_s+1} - x_{t,i_s}\| \right).$$

$$(15)$$

According to the above result, the challenge lies in the gap between $x_{\tau,i_s}$ and $x_{\tau,i_s+1}$.

LEMMA 7. *For each $t \in [T], k \in \mathcal{F}_t$, our DMD-GC algorithm ensures that*

$$\|x_{t,i_k+1} - x_{t,i_k}\| \leq \frac{\eta G_\star + 2\eta^2}{\sigma} + \frac{\eta^2 \xi G_\star}{\sigma^2}. \tag{16}$$

Substituting Eq. (16) into Eq. (15) gives

$$\sum_{t=1}^{T} \sum_{k \in \mathcal{F}_t} \left\langle \nabla f_k(x_k), x_k - x_{t,i_k} \right\rangle \leq 2D_T G_\star \left( \frac{\eta G_\star + 2\eta^2}{\sigma} + \frac{\eta^2 \xi G_\star}{\sigma^2} \right).$$

$$(17)$$

Combining Eq. (14) and Eq. (17) yields the result of Theorem 3. □

The regret bound specified in Theorem 3 depends on the distinctive properties of the regularization function $\psi$ and its domain. For instance, we provide the following examples with different norms.

COROLLARY 3. *Applying DMD-GC to Example 1 with $x_1 = \mathbf{0} \in \mathcal{X}$ and $\eta = \frac{R_2}{G_2} \sqrt{\frac{1}{T+4D_T}}$ gives $\text{Reg}_T \leq G_2 R_2 \sqrt{T + 4D_T} + 4\xi R_2^2 \sqrt{T} + 2\xi R_2^2 + \frac{5R_2^2}{G_2}$.*

*Applying DMD-GC to Example 2 with $\boldsymbol{x}_1 = [\frac{1}{n}, \ldots, \frac{1}{n}] \in \mathbb{R}^n_+$ and $\eta = \frac{1}{G_\infty}\sqrt{\frac{2\ln n}{T+4D_T}}$ gives $\mathrm{Reg}_T \le G_\infty\sqrt{2(T+4D_T)\ln n} + 4\xi\sqrt{2T\ln n} + 4\xi\ln n + \frac{10\ln n}{G_\infty}$.*

*Applying DMD-GC to Example 3 with $\boldsymbol{x}_1 = \mathbf{0} \in \mathcal{X}$ and $\eta = \frac{R_p}{G_q}\sqrt{\frac{p-1}{T+4D_T}}$ gives $\mathrm{Reg}_T \le R_p G_q\sqrt{\frac{T+4D_T}{p-1}} + 4\xi R_p^2\sqrt{\frac{T}{p-1}} + \frac{2\xi R_p^2}{p-1} + \frac{5R_p^2}{G_q}$.*

**Remark.** With access to gradient feedback, the regret of DMD-GC aligns with our FTDRL-GC which utilizes the full information of loss functions. If we assume $d_t = 1, \forall t \in [T]$, our results match the ones achieved in the standard non-delayed OMD algorithm [15].

## 5.2 Logarithmic Regret for Relative Strong Convexity

When dealing with functions that exhibit relative strong convexity, we introduce a new algorithm called Delayed Mirror Descent for Relative Strong Convexity (DMD-RSC) and outline its steps in Algorithm 4. There are two key differences between this algorithm and the one used for general convexity. Firstly, we perform a single mirror descent operation on the sum of gradients in the set $\{\nabla f_k : k \in \mathcal{F}_t\}$. Secondly, since a constant learning rate cannot take advantage of the relative strong convexity of loss functions, we use a decreasing learning rate $\eta_t = \frac{1}{\gamma\sum_{\tau=1}^t |\mathcal{F}_\tau|}$ that is determined by the total number of observable feedback.

We establish the following theorem regarding the regret bound.

**Theorem 4.** *Under Assumptions 1, 2, 3 and 4, let $\eta_t = \frac{1}{\gamma\sum_{\tau=1}^t |\mathcal{F}_\tau|}$ and maximum error $\rho_t = \frac{\eta_t^3}{2\sigma}, \forall t \in [T]$, Algorithm 4 satisfies*

$$\mathrm{Reg}_T \le \frac{(3dG_\star^2 + 8\xi RG_\star)(1+\ln T)}{2\sigma\gamma} + \frac{6dG_\star}{\sigma\gamma^2} + \frac{2d\xi G_\star^2}{\sigma^2\gamma^2}.$$

**Proof.** Here, the total regret is divided by

$$\mathrm{Reg}_T = \underbrace{\sum_{t=1}^T \sum_{k\in\mathcal{F}_t}[f_k(\boldsymbol{x}_t) - f_k(\boldsymbol{x}^*)]}_{\text{normal term}} + \underbrace{\sum_{t=1}^T \sum_{k\in\mathcal{F}_t}[f_k(\boldsymbol{x}_k) - f_k(\boldsymbol{x}_t)]}_{\text{delayed term}}. \tag{18}$$

For the normal term of Eq. (18), we have the following lemma.

**Lemma 8.** *The normal term of Rq. (18) is bounded by*

$$\sum_{t=1}^T \sum_{k\in\mathcal{F}_t}[f_k(\boldsymbol{x}_t) - f_k(\boldsymbol{x}^*)] \le \frac{(dG_\star^2 + 8\xi RG_\star)(1+\ln T)}{2\sigma\gamma} + \frac{2dG_\star}{\sigma\gamma^2}. \tag{19}$$

Next, we discuss the delayed term of Eq. (18).

$$\sum_{t=1}^T \sum_{k\in\mathcal{F}_t}[f_k(\boldsymbol{x}_k) - f_k(\boldsymbol{x}_t)] \le \sum_{t=1}^T \sum_{k\in\mathcal{F}_t}\sum_{\tau=k}^{t-1}[f_k(\boldsymbol{x}_\tau) - f_k(\boldsymbol{x}_{\tau+1})]$$

$$\le \sum_{t=1}^T \sum_{k\in\mathcal{F}_t}\sum_{\tau=k}^{k+d_k-1} G_\star\|\boldsymbol{x}_\tau - \boldsymbol{x}_{\tau+1}\| \le d\sum_{t=1}^T G_\star\|\boldsymbol{x}_t - \boldsymbol{x}_{t+1}\|. \tag{20}$$

The crucial step is to compute the gap between $\boldsymbol{x}_t$ and $\boldsymbol{x}_{t+1}$.

**Lemma 9.** *For each $t \in [T]$, our DMD-RSC algorithm ensures that*

$$\|\boldsymbol{x}_t - \boldsymbol{x}_{t+1}\| \le \frac{\eta_t|\mathcal{F}_t|G_\star + \eta_t^2 + \eta_{t-1}^2}{\sigma} + \frac{\eta_{t-1}^2\xi G_\star}{\sigma^2}. \tag{21}$$

Substituting Eq. (21) into Eq. (20) and making $\eta_t = \frac{1}{\sum_{\tau=1}^t |\mathcal{F}_\tau|\gamma}$ gives

$$\sum_{t=1}^T \sum_{k\in\mathcal{F}_t}[f_k(\boldsymbol{x}_k) - f_k(\boldsymbol{x}_t)] \le \frac{dG_\star^2(1+\ln T)}{\sigma\gamma} + \frac{4dG_\star}{\sigma\gamma^2} + \frac{2d\xi G_\star^2}{\sigma^2\gamma^2}. \tag{22}$$

Considering Eq. (19) and Eq. (22) together yields Theorem 4. □

To discuss the performance on the different norms, we give the following examples.

**Corollary 4.** *If we apply DMD-RSC to Example 1, we get $\mathrm{Reg}_T \le \frac{6dG_2}{\gamma^2} + \frac{2d\xi G_2^2}{\gamma^2} + \frac{(3dG_2^2+8\xi R_2G_2)(1+\ln T)}{2\gamma}$.*

*If we apply DMD-RSC to Example 2, we get $\mathrm{Reg}_T \le \frac{6dG_\infty}{\gamma^2} + \frac{2d\xi G_\infty^2}{\gamma^2} + \frac{(3dG_\infty^2+8\xi G_\infty)(1+\ln T)}{2\gamma}$.*

*If we apply DMD-RSC to Example 3, we get $\mathrm{Reg}_T \le \frac{6dG_q}{(1-p)\gamma^2} + \frac{2d\xi G_q^2}{(1-p)^2\gamma^2} + \frac{(3dG_q^2+8\xi R_pG_q)(1+\ln T)}{2\gamma(1-p)}$.*

**Remark.** With the feedback of gradient information, DMD-RSC algorithm produces matchable results to those of our FTDL-RSC. Furthermore, assuming $d_t = 1, \forall t \in [T]$, our findings align with the regret bound achieved in the non-delayed OMD algorithm [33] for relative strong convexity.

## 6 SIMPLIFIED DELAYED MIRROR DESCENT

In this section, the observable feedback reduces to the value information of the loss function's gradient at the corresponding decision point. In contrast to our proposed delayed mirror descent type algorithms utilizing the feedback of the gradient information, here we replace the gradient set $\{\nabla f_k : k \in \mathcal{F}_t\}$ with the set of gradients' values $\{\nabla f_k(\boldsymbol{x}_k) : k \in \mathcal{F}_t\}$. Additionally, the regret analysis follows a similar way as the previous section, and for brevity, we include it in the appendix.

## 6.1 Sublinear Regret for General Convexity

We propose a Simplified Delayed Mirror Descent for General Convexity (SDMD-GC) algorithm and outline its process in Algorithm 5. Similar to Algorithm 3, for each iteration $t$, we divide the update process into $|\mathcal{F}_t|$ segments of approximate solution and utilize a fixed learning rate.

We establish the following theorem to curve the regret bound.

**Theorem 5.** *Under Assumptions 1, 2 and 3, let the maximum approximate error $\rho_{t,i_k} = \frac{\eta^3}{2\sigma}, \forall k \in \mathcal{F}_t, t \in [T]$, Algorithm 5 satisfies*

$$\mathrm{Reg}_T \le \frac{\eta(G_\star^2 T + 8\xi RG_\star T + 2\eta G_\star T + 4G_\star^2 D_T + 8\eta G_\star D_T)}{2\sigma}$$

$$+ \frac{2\eta^2\xi G_\star^2 D_T}{\sigma^2} + \frac{B_\psi(\boldsymbol{x}^*; \boldsymbol{x}_1)}{\eta}.$$

**Algorithm 5** Simplified Delayed Mirror Descent for General Convexity

1: Input: both $\eta$ and $x_1$ depend on the choice of example
2: **for** $t = 1, \ldots, T$ **do**
3:     Query $\nabla f_t(x_t)$ and receive feedback $\{\nabla f_k(x_k) : k \in \mathcal{F}_t\}$
4:     **if** $|\mathcal{F}_t| > 0$ **then**
5:         $x_{t,0} = x_t$
6:         **for** $k \in \mathcal{F}_t$ **do**
7:           Make an approximate solution $x_{t,i_k+1}$:

$$\left\langle \nabla f_k(x_k), x_{t,i_k+1} \right\rangle + \frac{1}{\eta} B_\psi(x_{t,i_k+1}; x_{t,i_k})$$
$$\leq \left\langle \nabla f_k(x_k), y_{t,i_k+1}^* \right\rangle + \frac{1}{\eta} B_\psi(y_{t,i_k+1}^*; x_{t,i_k}) + \rho_{t,i_k},$$
$$\text{where } y_{t,i_k+1}^* = \arg\min_{x \in \mathcal{X}} \left\{ \left\langle \nabla f_k(x_k), x \right\rangle + \frac{1}{\eta} B_\psi(x; x_{t,i_k}) \right\}$$

8:         **end for**
9:         $x_{t+1} = x_{t,|\mathcal{F}_t|}$
10:    **else**
11:        $x_{t+1} = x_t$
12:    **end if**
13: **end for**

Note that the performance delineated in Theorem 5 is affected by the nature of the regularizer. In light of this, we provide the following examples.

COROLLARY 5. *Applying SDMD-GC to Example 1 with $x_1 = \mathbf{0} \in \mathcal{X}$, $\eta = \frac{R_2}{G_2}\sqrt{\frac{1}{T+4D_T}}$ gives $\text{Reg}_T \leq G_2 R_2 \sqrt{T + 4D_T} + 4\xi R_2^2 \sqrt{T} + 2\xi R_2^2 + \frac{5R_2^2}{G_2}$.*

*Applying SDMD-GC to Example 2 with $x_1 = [\frac{1}{n}, \ldots, \frac{1}{n}] \in \mathbb{R}_+^n$ and $\eta = \frac{1}{G_\infty}\sqrt{\frac{2\ln n}{T+4D_T}}$ gives $\text{Reg}_T \leq G_\infty \sqrt{2(T + 4D_T)\ln n} + 4\xi\sqrt{2T\ln n} + 4\xi \ln n + \frac{10\ln n}{G_\infty}$.*

*Apply SDMD-GC to Example 3 with $x_1 = \mathbf{0} \in \mathcal{X}$ and $\eta = \frac{R_p}{G_q}\sqrt{\frac{p-1}{T+4D_T}}$ gives $\text{Reg}_T \leq R_p G_q \sqrt{\frac{T+4D_T}{p-1}} + 4\xi R_p^2 \sqrt{\frac{T}{p-1}} + \frac{2\xi R_p^2}{p-1} + \frac{5R_p^2}{G_q}$.*

*Remark.* Despite having access to only the value information of gradient feedback at corresponding decision points instead of the full or gradient information of loss functions, we can still attain results consistent with those of FTDRL-GC and DMD-GC.

## 6.2 Logarithmic Regret for Relative Strong Convexity

In the case of relative strong convexity, we introduce an algorithm called Simplified Delayed Mirror Descent for Relative Strong Convexity (SDMD-RSC) and present its steps in Algorithm 6. Similar to Algorithm 4, we conduct a single approximate mirror descent operation on the joint sum of gradients in the set $\{\nabla f_k(x_k) : k \in \mathcal{F}_t\}$. However, there is a difference with Algorithm 4 in that the decreasing learning rate is not related to the amount of information observed but to the total number of decisions completed.

We establish the following theorem and corollary regarding the regret bound.

**Algorithm 6** Simplified Delayed Mirror Descent for Relative Strongly Convexity

1: Input: $\eta_t = \frac{1}{\gamma t}$, make an arbitrary decision $x_1 \in \mathcal{X}$
2: **for** $t = 1, \ldots, T$ **do**
3:     Query $\nabla f_t(x_t)$ and receive feedback $\{\nabla f_k(x_k) : k \in \mathcal{F}_t\}$
4:     **if** $|\mathcal{F}_t| > 0$ **then**
5:         Make an approximate solution $x_{t+1}$:

$$\left\langle \sum_{k \in \mathcal{F}_t} \nabla f_k(x_k), x_{t+1} \right\rangle + \frac{1}{\eta_t} B_\psi(x_{t+1}; x_t)$$
$$\leq \left\langle \sum_{k \in \mathcal{F}_t} \nabla f_k(x_k), y_{t+1}^* \right\rangle + \frac{1}{\eta_t} B_\psi(y_{t+1}^*; x_t) + \rho_t,$$
$$\text{where } y_{t+1}^* = \arg\min_{x \in \mathcal{X}} \left\{ \left\langle \sum_{k \in \mathcal{F}_t} \nabla f_k(x_k), x \right\rangle + \frac{1}{\eta_t} B_\psi(x; x_t) \right\}$$

6:    **else**
7:        $x_{t+1} = x_t$
8:    **end if**
9: **end for**

THEOREM 6. *Under Assumptions 1, 2, 3 and 4, let $\eta_t = \frac{1}{\gamma t}$ and the maximum error $\rho_t = \frac{\eta_t^3}{2\sigma}, \forall t \in [T]$, Algorithm 6 satisfies*

$$\text{Reg}_T \leq \frac{(3dG_\star^2 + 8\xi RG_\star)(1 + \ln T)}{2\sigma\gamma} + \frac{6dG_\star}{\sigma\gamma^2} + \frac{2d\xi G_\star}{\sigma^2\gamma^2}.$$

COROLLARY 6. *If we apply SDMD-RSC to Example 1, we get $\text{Reg}_T \leq \frac{6dG_2}{\gamma^2} + \frac{2d\xi G_2}{\gamma^2} + \frac{(3dG_2^2 + 8\xi R_2 G_2)(1 + \ln T)}{2\gamma}$.*

*If we apply SDMD-RSC to Example 2, we get $\text{Reg}_T \leq \frac{6dG_\infty}{\gamma^2} + \frac{2d\xi G_\infty}{\gamma^2} + \frac{(3dG_\infty^2 + 8\xi G_\infty)(1 + \ln T)}{2\gamma}$.*

*If we apply SDMD-RSC to Example 3, we get $\text{Reg}_T \leq \frac{6dG_q}{\gamma^2(1-p)} + \frac{2d\xi G_q}{\gamma^2(1-p)^2} + \frac{(3dG_q^2 + 8\xi R_p G_q)(1 + \ln T)}{2\gamma(1-p)}$.*

*Remark.* Considering relative strong convexity, even when only the values of the loss function's gradient at certain decision points are available, SDMD-RSC can achieve comparable results to our FTDL-RSC and DMD-RSC.

## 7 CONCLUSION

In the field of online sequential decision-making with unknown delays, we propose a range of learning algorithms, namely FTDRL, DMD and SDMD, to handle delayed full function information, full gradient information and value information of gradient at the decision point, respectively. Notably, our algorithms only necessitate an approximate solution for the optimization step and are applied to several cases, including general convexity and relative strong convexity, as well as specific examples utilizing different norms. Our next step is to explore more adaptable algorithms capable of simultaneously handling multiple types of loss functions and accommodating the flexibility of switching feedback types during the learning process.

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

# A  PREPARATION

## A.1  Proof of Lemma 3

Consider a specific term $\sum_{\tau=k}^{t-1} |\mathcal{F}_\tau| + |\mathcal{F}_{t,k}|$ within the sum. This term calculates the count of feedback instances, denoted by $\Xi_k$, that are withheld while other feedback is applied during iterations $k$ to $t \geq k$. Let's fix two iterations, $k$ and $t$, and consider an intermediate iteration $\tau \in \{k, \ldots, t\}$. If $\tau < t$, we fix $s \in \mathcal{F}_\tau$, and if $\tau = t$, we fix $s \in \mathcal{F}_{t,k}$. The feedback from iteration $s$ is applied during an iteration $\tau$ between $k$ and $t$. We divide our analysis into two scenarios: when $s \leq k$ and when $s > k$. In the second scenario, $s > k$, there are at most $d_k$ instances since $s$ must lie between $k+1$ and $t$. We can assign the first case to $d_s$. In the first scenario, the feedback from iteration $s$ appears only after $k$. We observe that for a fixed $s$, the number of indices $k$ such that $s < k \leq d_s + s \leq d_k + k$ is at most $d_s$. In other words, all instances of the second case for a fixed $s$ can be attributed to $d_s$. Between the two cases, we have

$$\sum_{t=1}^{T} \sum_{k \in \mathcal{F}_t} \left( \sum_{\tau=k}^{t-1} |\mathcal{F}_\tau| + |\mathcal{F}_{t,k}| \right) \leq \sum_{t=1}^{T} 2d_t \leq 2D_T.$$

# B  PROOF OF THEOREM 1

## B.1  Proof of Lemma 1

Call back the definition $\Phi_{t,i_k}(\boldsymbol{x}) = \sum_{\tau=1}^{t-1} \sum_{s \in \mathcal{F}_\tau} f_s(\boldsymbol{x}) + \sum_{s \in \mathcal{F}_{t,k}} f_s(\boldsymbol{x}) + f_k(\boldsymbol{x}) + \frac{1}{\eta} \psi(\boldsymbol{x})$ and $\Phi_0(\boldsymbol{x}) = \frac{1}{\eta} \psi(\boldsymbol{x})$.

For the normal term of Eq. (3), we have

$$\sum_{t=1}^{T} \sum_{k \in \mathcal{F}_t} [f_k(\boldsymbol{x}_{t,i_k}) - f_k(\boldsymbol{x}^*)] = \sum_{t=1}^{T} \sum_{k \in \mathcal{F}_t} [\Phi_{t,i_k}(\boldsymbol{x}_{t,i_k}) - \Phi_{t,i_k-1}(\boldsymbol{x}_{t,i_k})] - \sum_{t=1}^{T} \sum_{k \in \mathcal{F}_t} f_k(\boldsymbol{x}^*). \tag{23}$$

Due to the fact that $\boldsymbol{y}_{T+1}^*$ (i.e., $\boldsymbol{y}_{T,|\mathcal{F}_T|}^*$) is the minimizer of $\Phi_{T,|\mathcal{F}_T|-1}(\boldsymbol{x})$ over $\boldsymbol{x} \in \mathcal{X}$, thus

$$\Phi_{T,|\mathcal{F}_T|-1}(\boldsymbol{y}_{T+1}^*) \leq \sum_{t=1}^{T} \sum_{k \in \mathcal{F}_t} f_k(\boldsymbol{x}^*) + \frac{1}{\eta} \psi(\boldsymbol{x}^*). \tag{24}$$

Combining with Eq. (23) and Eq. (24) gives

$$\begin{aligned}
\sum_{t=1}^{T} \sum_{k \in \mathcal{F}_t} [f_k(\boldsymbol{x}_{t,i_k}) - f_k(\boldsymbol{x}^*)] &\leq \sum_{t=1}^{T} \sum_{k \in \mathcal{F}_t} [\Phi_{t,i_k}(\boldsymbol{x}_{t,i_k}) - \Phi_{t,i_k-1}(\boldsymbol{x}_{t,i_k})] - \Phi_{T,|\mathcal{F}_T|-1}(\boldsymbol{y}_{T+1}^*) + \frac{1}{\eta} \psi(\boldsymbol{x}^*) \\
&= \sum_{t=1}^{T} \sum_{k \in \mathcal{F}_t} [\Phi_{t,i_k}(\boldsymbol{y}_{t,i_k}^*) - \Phi_{t,i_k-1}(\boldsymbol{y}_{t,i_k}^*)] - \Phi_{T,|\mathcal{F}_T|-1}(\boldsymbol{y}_{T+1}^*) + \frac{1}{\eta} \psi(\boldsymbol{x}^*) \\
&\quad + \sum_{t=1}^{T} \sum_{k \in \mathcal{F}_t} [\Phi_{t,i_k}(\boldsymbol{x}_{t,i_k}) - \Phi_{t,i_k-1}(\boldsymbol{x}_{t,i_k}) - \Phi_{t,i_k}(\boldsymbol{y}_{t,i_k}^*) + \Phi_{t,i_k-1}(\boldsymbol{y}_{t,i_k}^*)] \\
&= \sum_{t=1}^{T} \sum_{k \in \mathcal{F}_t} [\Phi_{t,i_k}(\boldsymbol{y}_{t,i_k}^*) - \Phi_{t,i_k-1}(\boldsymbol{y}_{t,i_k}^*)] - \Phi_{T,|\mathcal{F}_T|-1}(\boldsymbol{y}_{T+1}^*) + \frac{1}{\eta} \psi(\boldsymbol{x}^*) \\
&\quad + \sum_{t=1}^{T} \sum_{k \in \mathcal{F}_t} [f_k(\boldsymbol{x}_{t,i_k}) - f_k(\boldsymbol{y}_{t,i_k}^*)] \\
&\leq \sum_{t=1}^{T} \sum_{k \in \mathcal{F}_t} [\Phi_{t,i_k}(\boldsymbol{y}_{t,i_k}^*) - \Phi_{t,i_k}(\boldsymbol{y}_{t,i_k+1}^*)] + \frac{1}{\eta} [\psi(\boldsymbol{x}^*) - \psi(\boldsymbol{x}_1)] \\
&\quad + \sum_{t=1}^{T} \sum_{k \in \mathcal{F}_t} G_\star \|\boldsymbol{x}_{t,i_k} - \boldsymbol{y}_{t,i_k}^*\|.
\end{aligned} \tag{25}$$

For the first term on the R.H.S of the above formula, we have

$$
\begin{aligned}
\Phi_{t,i_k}(\boldsymbol{y}^*_{t,i_k}) - \Phi_{t,i_k}(\boldsymbol{y}^*_{t,i_k+1}) =& \Phi_{t,i_k-1}(\boldsymbol{y}^*_{t,i_k}) - \Phi_{t,i_k-1}(\boldsymbol{y}^*_{t,i_k+1}) + f_k(\boldsymbol{y}^*_{t,i_k}) - f_k(\boldsymbol{y}^*_{t,i_k+1}) \\
\leq& \langle \nabla\Phi_{t,i_k-1}(\boldsymbol{y}^*_{t,i_k}), \boldsymbol{y}^*_{t,i_k} - \boldsymbol{y}^*_{t,i_k+1} \rangle - \frac{\sigma}{2\eta}\|\boldsymbol{y}^*_{t,i_k} - \boldsymbol{y}^*_{t,i_k+1}\|^2 \\
&+ \langle \nabla f_k(\boldsymbol{y}^*_{t,i_k}), \boldsymbol{y}^*_{t,i_k} - \boldsymbol{y}^*_{t,i_k+1} \rangle \\
\leq& \frac{\eta}{2\sigma}\|\nabla f_k(\boldsymbol{y}^*_{t,i_k})\|^2_\star + \frac{\sigma}{2\eta}\|\boldsymbol{y}^*_{t,i_k} - \boldsymbol{y}^*_{t,i_k+1}\|^2 - \frac{\sigma}{2\eta}\|\boldsymbol{y}^*_{t,i_k} - \boldsymbol{y}^*_{t,i_k+1}\|^2 \\
\leq& \frac{\eta}{2\sigma}G^2_\star.
\end{aligned}
\tag{26}
$$

The first inequality is due to the fact that $\psi$ is $\sigma$-strongly convex with respect to a norm $\|\cdot\|$. The second inequality is due to the optimality condition of the update rule (i.e., $\boldsymbol{x}^* = \arg\min_{\boldsymbol{x}\in\mathcal{X}} f(\boldsymbol{x})$ iff $\langle \nabla f(\boldsymbol{x}^*), \boldsymbol{y} - \boldsymbol{x}^* \rangle \geq 0, \forall \boldsymbol{y} \in \mathcal{X}$) and $\langle \boldsymbol{u}, \boldsymbol{v} \rangle \leq \frac{\boldsymbol{u}^2}{2\alpha} + \frac{\alpha\boldsymbol{v}^2}{2}$ for any $\boldsymbol{u}, \boldsymbol{v} \in \mathcal{X}, \alpha > 0$.

Considering the last term of Eq. (25) which means the error incurred by the approximate solution, we have

$$
\Phi_{t,i_k}(\boldsymbol{x}_{t,i_k+1}) \leq \Phi_{t,i_k}(\boldsymbol{y}^*_{t,i_k+1}) + \rho_{t,i_k}.
\tag{27}
$$

Meanwhile, we utilize the fact that $\Phi_{t,i_k}$ is $\frac{\sigma}{\eta}$-strongly convex with respect to norm $\|\cdot\|$ to obtain

$$
\begin{aligned}
\Phi_{t,i_k}(\boldsymbol{x}_{t,i_k+1}) - \Phi_{t,i_k}(\boldsymbol{y}^*_{t,i_k+1}) \geq& \langle \nabla\Phi_{t,i_k}(\boldsymbol{y}^*_{t,i_k+1}), \boldsymbol{x}_{t,i_k+1} - \boldsymbol{y}^*_{t,i_k+1} \rangle + \frac{\sigma}{2\eta}\|\boldsymbol{x}_{t,i_k+1} - \boldsymbol{y}^*_{t,i_k+1}\|^2 \\
\geq& \frac{\sigma}{2\eta}\|\boldsymbol{x}_{t,i_k+1} - \boldsymbol{y}^*_{t,i_k+1}\|^2.
\end{aligned}
\tag{28}
$$

The last inequality is due to the optimality condition.

Consider $\rho_{t,i_k} = \frac{\eta G^2_\star}{8\sigma}$, combining with Eq. (27) and Eq. (28) yields

$$
\|\boldsymbol{x}_{t,i_k+1} - \boldsymbol{y}^*_{t,i_k+1}\| \leq \sqrt{\frac{2\eta\rho_{t,i_k}}{\sigma}} = \frac{\eta G_\star}{2\sigma}.
\tag{29}
$$

Substituting Eq. (26) and Eq. (29) into Eq. (25) yields the result of Lemma 1.

## B.2 Proof of Lemma 2

Due to the fact that $f_t$ is convex and $\psi$ is $\sigma$-strongly convex with respect to norm $\|\cdot\|$, we have

$$
\begin{aligned}
\Phi_{t,i_k}(\boldsymbol{y}^*_{t,i_k}) - \Phi_{t,i_k}(\boldsymbol{y}^*_{t,i_k+1}) \geq& \langle \nabla\Phi_{t,i_k}(\boldsymbol{y}^*_{t,i_k+1}), \boldsymbol{y}^*_{t,i_k} - \boldsymbol{y}^*_{t,i_k+1} \rangle + \frac{\sigma}{2\eta}\|\boldsymbol{y}^*_{t,i_k} - \boldsymbol{y}^*_{t,i_k+1}\|^2 \\
\geq& \frac{\sigma}{2\eta}\|\boldsymbol{y}^*_{t,i_k} - \boldsymbol{y}^*_{t,i_k+1}\|^2.
\end{aligned}
\tag{30}
$$

The last inequality is due to the fact that $\boldsymbol{y}^*_{t,i_k+1}$ is the minimizer of $\Phi_{t,i_k}(\boldsymbol{x})$ over $\boldsymbol{x} \in \mathcal{X}$.

Meanwhile, the L.H.S of the above formula is upper bounded by

$$
\begin{aligned}
\Phi_{t,i_k}(\boldsymbol{y}^*_{t,i_k}) - \Phi_{t,i_k}(\boldsymbol{y}^*_{t,i_k+1}) =& \Phi_{t,i_k-1}(\boldsymbol{y}^*_{t,i_k}) - \Phi_{t,i_k-1}(\boldsymbol{y}^*_{t,i_k+1}) + f_k(\boldsymbol{y}^*_{t,i_k}) - f_k(\boldsymbol{y}^*_{t,i_k+1}) \\
\leq& \langle \nabla\Phi_{t,i_k-1}(\boldsymbol{y}^*_{t,i_k}), \boldsymbol{y}^*_{t,i_k} - \boldsymbol{y}^*_{t,i_k+1} \rangle - \frac{\sigma}{2\eta}\|\boldsymbol{y}^*_{t,i_k} - \boldsymbol{y}^*_{t,i_k+1}\|^2 \\
&+ G_\star\|\boldsymbol{y}^*_{t,i_k} - \boldsymbol{y}^*_{t,i_k+1}\| \\
\leq& G_\star\|\boldsymbol{y}^*_{t,i_k} - \boldsymbol{y}^*_{t,i_k+1}\| - \frac{\sigma}{2\eta}\|\boldsymbol{y}^*_{t,i_k} - \boldsymbol{y}^*_{t,i_k+1}\|^2.
\end{aligned}
\tag{31}
$$

The last inequality is due to the fact that $\boldsymbol{y}^*_{t,i_k}$ is the minimizer of $\Phi_{t,i_k-1}(\boldsymbol{x})$ over $\boldsymbol{x} \in \mathcal{X}$.

Combining Eq. (30) and Eq. (31) gives

$$
\|\boldsymbol{y}^*_{t,i_k+1} - \boldsymbol{y}^*_{t,i_k}\| \leq \frac{\eta G_\star}{\sigma}.
$$

Then, utilizing the result of Eq. (29), we have

$$
\begin{aligned}
\|\boldsymbol{x}_{t,i_k+1} - \boldsymbol{x}_{t,i_k}\| \leq& \|\boldsymbol{y}^*_{t,i_k+1} - \boldsymbol{y}^*_{t,i_k}\| + \|\boldsymbol{x}_{t,i_k+1} - \boldsymbol{y}^*_{t,i_k+1}\| + \|\boldsymbol{x}_{t,i_k} - \boldsymbol{y}^*_{t,i_k}\| \\
\leq& \frac{2\eta G_\star}{\sigma}.
\end{aligned}
$$

## B.3 Proof of Corollary 1

*Example 1 of FTDRL-GC.* In the Euclidean space, we first preset $x_1 = 0^n$ and make $\psi(x) = \frac{1}{2}\|x\|_2^2$. Note that the dual norm of $\|\cdot\|_2$ is itself and $B_\psi(x; y) = \frac{1}{2}\|x - y\|_2^2$. Additionally, we assume that $\|\nabla f_t(x)\|_2 \le G_2$ and $\|x\| \le R_2$ for any $x \in \mathcal{X}, t \in [T]$.

Based on the result of Theorem 1, we can get the following regret

$$\text{Reg}_T \le \frac{\eta}{\sigma} T G_2^2 + \frac{4\eta}{\sigma} D_T G_2^2 + \frac{R_2^2}{2\eta}$$

$$= \eta T G_2^2 + 4\eta D_T G_2^2 + \frac{R_2^2}{2\eta}.$$

The last equality is because $\psi(x) = \frac{1}{2}\|x\|_2^2$ is 1-strongly convex with respect to norm $\|\cdot\|_2$.

To minimize above, we make $\eta = \frac{R_2}{G_2}\sqrt{\frac{1}{2T+8D_T}}$, thus

$$\text{Reg}_T \le G_2 R_2 \sqrt{2T + 8D_T}.$$

*Example 2 of FTDRL-GC.* In the probabilistic simplex, the convex set $\mathcal{X} = \{x \in \mathbb{R}_+^n : \|x\|_1 = 1\}$. We set $\psi(x) = \sum_{\mu=1}^n x^{(\mu)} \ln x^{(\mu)} + \ln n$ and the initial decision $x_1 = [\frac{1}{n}, \ldots, \frac{1}{n}] \in \mathbb{R}_+^n$. The dual norm of $\|\cdot\|_1$ is $\|\cdot\|_\infty$. Note that $\psi(x)$ is 1-strongly convex with respect to norm $\|\cdot\|_1$. Additionally, we assume that $\|\nabla f_t(x)\|_\infty \le G_\infty$ for any $x \in \mathcal{X}, t \in [T]$.

Considering $\psi(x^*) \le \ln n$ and $\psi(x_1) = 0$, we have

$$\text{Reg}_T \le \frac{\eta}{\sigma} T G_\infty^2 + \frac{4\eta}{\sigma} D_T G_\infty^2 + \frac{\ln n}{\eta}$$

$$= \eta T G_\infty^2 + 4\eta D_T G_\infty^2 + \frac{\ln n}{\eta}.$$

The last equality is due to the fact that $\psi(x)$ is 1-strongly convex with respect to norm $\|\cdot\|_1$, i.e., $\sigma = 1$.

To minimize above, we set $\eta = \frac{1}{G_\infty}\sqrt{\frac{\ln n}{T+4D_T}}$, then

$$\text{Reg}_T \le 2 G_\infty \sqrt{(T + 4D_T) \ln n}.$$

*Example 3 of FTDRL-GC.* Consider the regularization function $\psi(x) = \frac{1}{2}\|x\|_p^2$, where $\|x\|_p = \left(\sum_{\mu=1}^n |x^{(\mu)}|^p\right)^{\frac{1}{p}}$ and $1 < p \le 2$ over $\mathcal{X} \in \mathbb{R}^n$. We preset $x_1 = 0^n$. The dual norm of $\|\cdot\|_p$ is $\|\cdot\|_q$, where $\frac{1}{p} + \frac{1}{q} = 1$. We assume that $\|\nabla f_t(x)\|_q \le G_q$ and $\|x\|_p \le R_p$ for any $x \in \mathcal{X}, t \in [T]$.

Here the regret bound is

$$\text{Reg}_T \le \frac{\eta}{\sigma} T G_q^2 + \frac{4\eta}{\sigma} D_T G_q^2 + \frac{R_p^2}{2\eta}$$

$$= \frac{\eta}{p-1} T G_q^2 + \frac{4\eta}{p-1} D_T G_q^2 + \frac{R_p^2}{2\eta}.$$

The last equality is due to the fact that $\psi(x)$ is $(p-1)$-strongly convex with respect to norm $\|\cdot\|_p$, i.e., $\sigma = p - 1$.

To minimize the regret, we make $\eta = \frac{R_p}{G_q}\sqrt{\frac{p-1}{2T+8D_T}}$, thus

$$\text{Reg}_T \le R_p G_q \sqrt{\frac{2T + 8D_T}{p-1}}.$$

## C PROOF OF THEOREM 2

### C.1 Proof of Lemma 4

In the FTDL-RSC algorithm, the decision is updated by

$$y_{t+1}^* = \arg\min_{x \in \mathcal{X}} \left\{ \sum_{\tau=1}^t \sum_{k \in \mathcal{F}_\tau} f_k(x) \right\}$$

and the approximate solution $x_{t+1}$ satisfies

$$\sum_{\tau=1}^t \sum_{k \in \mathcal{F}_\tau} f_k(x_{t+1}) \le \sum_{\tau=1}^t \sum_{k \in \mathcal{F}_\tau} f_k(y_{t+1}^*) + \rho_t.$$

For convenience, we make $F_t(x) = \sum_{\tau=1}^t \sum_{k \in \mathcal{F}_\tau} f_k(x)$ and $F_0(x) = 0$, due to the strong convexity assumption of the loss function, $F_t(x)$ is $\sum_{\tau=1}^t |\mathcal{F}_\tau| \gamma$-strongly convex relative to function $\psi$.

For the normal term of Eq. (8), we have

$$\sum_{t=1}^T \sum_{k \in \mathcal{F}_t} f_k(x_t) - \sum_{t=1}^T \sum_{k \in \mathcal{F}_t} f_k(x^*) = \sum_{t=1}^T [F_t(x_t) - F_{t-1}(x_t)] - F_T(x^*).$$

Due to the fact that $y_{T+1}^*$ is the minimizer of $F_T(x)$, then $F_T(y_{T+1}^*) \leq F_T(x^*)$, substituting it into the above formula, we have

$$\sum_{t=1}^T \sum_{k \in \mathcal{F}_t} f_k(x_t) - \sum_{t=1}^T \sum_{k \in \mathcal{F}_t} f_k(x^*) \leq \sum_{t=1}^T [F_t(x_t) - F_{t-1}(x_t)] - F_T(y_{T+1}^*)$$

$$= \sum_{t=1}^T [F_t(y_t^*) - F_{t-1}(y_t^*)] - F_T(y_{T+1}^*)$$

$$+ \sum_{t=1}^T [F_t(x_t) - F_{t-1}(x_t) - F_t(y_t^*) + F_{t-1}(y_t^*)] \tag{32}$$

$$= \sum_{t=1}^T [F_t(y_t^*) - F_t(y_{t+1}^*)] - F_0(y_1^*) + \sum_{t=1}^T \sum_{k \in \mathcal{F}_t} [f_k(x_t) - f_k(y_t^*)]$$

$$\leq \sum_{t=1}^T [F_t(y_t^*) - F_t(y_{t+1}^*)] + \sum_{t=1}^T |\mathcal{F}_t| G_\star \|x_t - y_t^*\|.$$

Considering the optimality condition of the update rule and the relative strong convexity of loss functions, we have

$$F_t(y_t^*) - F_t(y_{t+1}^*) = F_{t-1}(y_t^*) - F_{t-1}(y_{t+1}^*) + \sum_{k \in \mathcal{F}_t} [f_k(y_t^*) - f_k(y_{t+1}^*)]$$

$$\leq \langle \nabla F_{t-1}(y_t^*), y_t^* - y_{t+1}^* \rangle - \sum_{\tau=1}^{t-1} \gamma |\mathcal{F}_\tau| B_\psi(y_{t+1}^*; y_t^*)$$

$$+ |\mathcal{F}_t| G_\star \cdot \|y_t^* - y_{t+1}^*\| - \gamma |\mathcal{F}_t| B_\psi(y_{t+1}^*; y_t^*)$$

$$\leq |\mathcal{F}_t| G_\star \cdot \|y_t^* - y_{t+1}^*\| - \sum_{\tau=1}^t \gamma |\mathcal{F}_\tau| B_\psi(y_{t+1}^*; y_t^*) \tag{33}$$

$$\leq |\mathcal{F}_t| G_\star \cdot \|y_t^* - y_{t+1}^*\| - \sum_{\tau=1}^t \frac{\sigma \gamma |\mathcal{F}_\tau|}{2} \|y_t^* - y_{t+1}^*\|^2$$

$$\leq \frac{|\mathcal{F}_t|^2 G_\star^2}{\sum_{\tau=1}^t 2\sigma \gamma |\mathcal{F}_\tau|}.$$

The third inequality is due to the fact that $B_\psi(y_{t+1}^*; y_t^*) \geq \frac{\sigma}{2} \|y_t^* - y_{t+1}^*\|$ if $\psi$ is $\sigma$-strongly convex with respect to norm $\|\cdot\|$. The last inequality is because $\langle u, v \rangle \leq \frac{\|u\|^2}{2\alpha} + \frac{\alpha \|v\|_\star^2}{2}$ for any $u, v \in \mathcal{X}, \alpha > 0$.

Considering the approximate solution gives

$$F_t(x_{t+1}) \leq F_t(y_{t+1}^*) + \rho_t \tag{34}$$

Due to $F_t$ is $\sum_{\tau=1}^t |\mathcal{F}_\tau| \gamma$-strongly convex relative to $\psi$, we obtain

$$F_t(x_{t+1}) - F_t(y_{t+1}^*) \geq \langle \nabla F_t(y_{t+1}^*), x_{t+1} - y_{t+1}^* \rangle + \sum_{\tau=1}^t |\mathcal{F}_\tau| \gamma B_\psi(x_{t+1}; y_{t+1}^*)$$

$$\geq \sum_{\tau=1}^t \frac{|\mathcal{F}_\tau| \gamma \sigma}{2} \|x_{t+1} - y_{t+1}^*\|^2 \tag{35}$$

The above inequality is due to the optimality condition and $\psi$ is $\sigma$-strongly convex with respect to norm $\|\cdot\|$.

Combining with Eq. (34) and Eq. (35) gives

$$\|x_{t+1} - y_{t+1}^*\| \leq \sqrt{\frac{2\rho_t}{\sum_{\tau=1}^t |\mathcal{F}_\tau| \gamma \sigma}}. \tag{36}$$

Considering $|\mathcal{F}_t| \le d, \forall t \in [T]$ and substituting Eq. (33) and Eq. (36) into Eq. (32) yields

$$\sum_{t=1}^{T} \sum_{k \in \mathcal{F}_t} f_k(\boldsymbol{x}_t) - \sum_{t=1}^{T} \sum_{k \in \mathcal{F}_t} f_k(\boldsymbol{x}^*) \le \frac{d|\mathcal{F}_t|G_\star^2}{\sum_{\tau=1}^{t} 2\sigma\gamma|\mathcal{F}_\tau|} + dG_\star \sqrt{\frac{2\rho_t}{\sum_{\tau=1}^{t}|\mathcal{F}_\tau|\gamma\sigma}}.$$

Note that $\sum_{t=1}^{T} \frac{|\mathcal{F}_t|}{\sum_{\tau=1}^{t}|\mathcal{F}_\tau|} \le 1 + \ln T$ and make $\rho_t = \frac{|\mathcal{F}_t|^2 G_\star^2}{8\sum_{\tau=1}^{t}|\mathcal{F}_\tau|\gamma\sigma}$. Then we get the result of Lemma 4 by combining Eq. (32) and Eq. (33).

$$
\begin{aligned}
\sum_{t=1}^{T} \sum_{k \in \mathcal{F}_t} [f_k(\boldsymbol{x}_t) - f_k(\boldsymbol{x}^*)] &\le \sum_{t=1}^{T} \frac{d|\mathcal{F}_t|G_\star^2}{\sum_{\tau=1}^{t} 2\sigma\gamma|\mathcal{F}_\tau|} + \sum_{t=1}^{T} \frac{d|\mathcal{F}_t|G_\star^2}{\sum_{\tau=1}^{t} 2\sigma\gamma|\mathcal{F}_\tau|} \\
&\le \frac{dG_\star^2}{\sigma\gamma}(1 + \ln T).
\end{aligned}
\tag{37}
$$

## C.2 Proof of Lemma 5

Combining with the relative strong convexity and the optimality condition of the update rule, we obtain

$$
\begin{aligned}
F_t(\boldsymbol{y}_t^*) - F_t(\boldsymbol{y}_{t+1}^*) &\ge \langle \nabla F_t(\boldsymbol{y}_{t+1}^*), \boldsymbol{y}_t^* - \boldsymbol{y}_{t+1}^* \rangle + \sum_{\tau=1}^{t} |\mathcal{F}_\tau|\gamma B_\psi(\boldsymbol{y}_t^*; \boldsymbol{y}_{t+1}^*) \\
&\ge \sum_{\tau=1}^{t} |\mathcal{F}_\tau|\gamma B_\psi(\boldsymbol{y}_t^*; \boldsymbol{y}_{t+1}^*) \\
&\ge \sum_{\tau=1}^{t} \frac{|\mathcal{F}_\tau|\gamma\sigma}{2} \|\boldsymbol{y}_t^* - \boldsymbol{y}_{t+1}^*\|^2.
\end{aligned}
\tag{38}
$$

Meanwhile, we have

$$
\begin{aligned}
F_t(\boldsymbol{y}_t^*) - F_t(\boldsymbol{y}_{t+1}^*) =& F_{t-1}(\boldsymbol{y}_t^*) - F_{t-1}(\boldsymbol{y}_{t+1}^*) + \sum_{k \in \mathcal{F}_t} [f_k(\boldsymbol{y}_t^*) - f_k(\boldsymbol{y}_{t+1}^*)] \\
\le& \langle \nabla F_{t-1}(\boldsymbol{y}_t^*), \boldsymbol{y}_t^* - \boldsymbol{y}_{t+1}^* \rangle + \sum_{k \in \mathcal{F}_t} \langle \nabla f_k(\boldsymbol{y}_t^*), \boldsymbol{y}_t^* - \boldsymbol{y}_{t+1}^* \rangle \\
& - \sum_{\tau=1}^{t} \gamma|\mathcal{F}_\tau|B_\psi(\boldsymbol{y}_{t+1}^*; \boldsymbol{y}_t^*) \\
\le& |\mathcal{F}_t|G_\star \|\boldsymbol{y}_t^* - \boldsymbol{y}_{t+1}^*\| - \sum_{\tau=1}^{t} \frac{\gamma\sigma|\mathcal{F}_\tau|}{2} \|\boldsymbol{y}_t^* - \boldsymbol{y}_{t+1}^*\|^2.
\end{aligned}
\tag{39}
$$

The first inequality is due to the relative strong convexity of loss functions. The last inequality is derived from the strong convexity of $\psi$ and Lipschitz continuity of loss functions.

Combining with Eq. (38) and Eq. (39) gives

$$\|\boldsymbol{y}_{t+1}^* - \boldsymbol{y}_t^*\| \le \frac{|\mathcal{F}_t|G_\star}{\sum_{\tau=1}^{t}|\mathcal{F}_\tau|\gamma\sigma}.$$

Utilizing the result of Eq. (36), we obtain

$$
\begin{aligned}
\|\boldsymbol{x}_{t+1} - \boldsymbol{x}_t\| \le& \|\boldsymbol{y}_{t+1}^* - \boldsymbol{y}_t^*\| + \|\boldsymbol{x}_{t+1} - \boldsymbol{y}_{t+1}^*\| + \|\boldsymbol{x}_t - \boldsymbol{y}_t^*\| \\
=& \frac{3}{2} \frac{|\mathcal{F}_t|G_\star}{\sum_{\tau=1}^{t}|\mathcal{F}_\tau|\gamma\sigma} + \frac{1}{2} \frac{|\mathcal{F}_{t-1}|G_\star}{\sum_{\tau=1}^{t-1}|\mathcal{F}_\tau|\gamma\sigma}.
\end{aligned}
$$

## C.3 Proof of Corollary 2

*Example 1 of FTDL-RSC.* In the Euclidean space, we first preset $\boldsymbol{x}_1 = \boldsymbol{0}^n$ and make $\psi(\boldsymbol{x}) = \frac{1}{2}\|\boldsymbol{x}\|_2^2$. Note that the dual norm of $\|\cdot\|_2$ is itself and $B_\psi(\boldsymbol{x}; \boldsymbol{y}) = \frac{1}{2}\|\boldsymbol{x} - \boldsymbol{y}\|_2^2$. Additionally, we assume that $\|\nabla f_t(\boldsymbol{x})\|_2 \le G_2$ for any $\boldsymbol{x} \in \mathcal{X}, t \in [T]$. Specially, $\psi(\boldsymbol{x}) = \frac{1}{2}\|\boldsymbol{x}\|_2^2$ is 1-strongly convex with respect to norm $\|\cdot\|_2$, i.e., $\sigma = 1$.

Thus, we have

$$\text{Reg}_T \le \frac{3dG_2^2}{\gamma}(1 + \ln T).$$

*Example 2 of FTDL-RSC.* In the probabilistic simplex, the convex set $X = \{x \in \mathbb{R}^n_+ : \|x\|_1 = 1\}$. We set $\psi(x) = \sum_{\mu=1}^{n} x^{(\mu)} \ln x^{(\mu)} + \ln n$ and the initial decision $x_1 = [\frac{1}{n}, \ldots, \frac{1}{n}] \in \mathbb{R}^n_+$. The dual norm of $\|\cdot\|_1$ is $\|\cdot\|_\infty$. Note that $\psi(x)$ is 1-strongly convex with respect to norm $\|\cdot\|_1$, i.e, $\sigma = 1$. Additionally, we assume that $\|\nabla f_t(x)\|_\infty \leq G_\infty$ for any $x \in X, t \in [T]$.

The regret is bounded by

$$\text{Reg}_T \leq \frac{3dG_\infty^2}{\gamma}(1 + \ln T).$$

The last inequality is due to the fact that $\psi(x)$ is 1-strongly convex with respect to norm $\|\cdot\|_1$, i.e., $\sigma = 1$.

*Example 3 of FTDL-RSC.* Consider the regularization function $\psi(x) = \frac{1}{2}\|x\|_p^2$, where $\|x\|_p = \left(\sum_{\mu=1}^n |x^{(\mu)}|^p\right)^{\frac{1}{p}}$ and $1 < p \leq 2$ over $X \in \mathbb{R}^n$. We preset $x_1 = 0^n$. The dual norm of $\|\cdot\|_p$ is $\|\cdot\|_q$, where $\frac{1}{p} + \frac{1}{q} = 1$. We assume that $\|\nabla f_t(x)\|_q \leq G_q$ and $\|x\|_p \leq R_p$ for any $x \in X, t \in [T]$.

The crucial ingredient [29] is the fact that $\psi(x)$ is $(p-1)$ strongly convex with respect to norm $\|\cdot\|_p$, i.e., $\sigma = p - 1$. Then, we have the result as follows:

$$\text{Reg}_T \leq \frac{3dG_q^2}{\gamma(p-1)}(1 + \ln T).$$

The proof of the corollary in the subsequent section is analogous to the process outlined in this section, therefore, we omit the proof of the subsequent corollary.

# D  PROOF OF THEOREM 3

## D.1  Proof of Lemma 6

The decision $y^*_{t,i_k+1}$ in DMD-GC algorithm is updated by

$$y^*_{t,i_k+1} = \arg\min_{x \in X} \left\{ \langle \nabla f_k(x_{t,i_k}), x \rangle + \frac{1}{\eta} B_\psi(x; x_{t,i_k}) \right\}.$$

According to the optimality condition of the update rule, for any $x^* \in X$, we have

$$\left\langle \eta \nabla f_k(x_{t,i_k}) + \nabla \psi(y^*_{t,i_k+1}) - \nabla \psi(x_{t,i_k}), x^* - y^*_{t,i_k+1} \right\rangle \geq 0.$$

Under Assumptions 1 and 2, rearranging the terms above yields

$$\langle \nabla f_k(x_{t,i_k}), x_{t,i_k} - x^* \rangle$$
$$\leq \langle \nabla f_k(x_{t,i_k}), x_{t,i_k} - y^*_{t,i_k+1} \rangle + \frac{\langle \nabla \psi(y^*_{t,i_k+1}) - \nabla \psi(x_{t,i_k}), x^* - y^*_{t,i_k+1} \rangle}{\eta}$$
$$\leq \langle \nabla f_k(x_{t,i_k}), x_{t,i_k} - x_{t,i_k+1} \rangle + \langle \nabla f_k(x_{t,i_k}), x_{t,i_k+1} - y^*_{t,i_k+1} \rangle$$
$$+ \frac{\langle \nabla \psi(x_{t,i_k+1}) - \nabla \psi(x_{t,i_k}), x^* - x_{t,i_k+1} \rangle}{\eta} + \frac{\langle \nabla \psi(y^*_{t,i_k+1}) - \nabla \psi(x_{t,i_k+1}), x^* - y^*_{t,i_k+1} \rangle}{\eta} \tag{40}$$
$$+ \frac{\langle \nabla \psi(x_{t,i_k+1}) - \nabla \psi(x_{t,i_k}), x_{t,i_k+1} - y^*_{t,i_k+1} \rangle}{\eta}$$
$$\leq \langle \nabla f_k(x_{t,i_k}), x_{t,i_k} - x_{t,i_k+1} \rangle + G_\star \|x_{t,i_k+1} - y^*_{t,i_k+1}\|$$
$$+ \frac{\langle \nabla \psi(x_{t,i_k+1}) - \nabla \psi(x_{t,i_k}), x^* - x_{t,i_k+1} \rangle}{\eta} + \frac{4\xi R G_\star \|x_{t,i_k+1} - y^*_{t,i_k+1}\|}{\eta}.$$

An interesting and useful identity regarding Bregman divergence, sometimes called three-point identity [9], is

$$\left\langle \nabla \psi(x_{t,i_k+1}) - \nabla \psi(x_{t,i_k}), x^* - x_{t,i_k+1} \right\rangle = B_\psi(x^*; x_{t,i_k}) - B_\psi(x^*; x_{t,i_k+1}) - B_\psi(x_{t,i_k+1}; x_{t,i_k}).$$

For the first term on the R.H.S of Eq.(40), we have

$$\langle \nabla f_k(x_{t,i_k}), x_{t,i_k} - x_{t,i_k+1} \rangle \leq G_\star \|x_{t,i_k} - x_{t,i_k+1}\|$$
$$\leq \frac{\eta}{2\sigma} G_\star^2 + \frac{\sigma}{2\eta} \|x_{t,i_k} - x_{t,i_k+1}\|^2 \tag{41}$$
$$\leq \frac{\eta}{2\sigma} G_\star^2 + \frac{B_\psi(x_{t,i_k+1}; x_{t,i_k})}{\eta}.$$

The last inequality is due to the regularization function $\psi(\cdot)$ is $\sigma$-strongly convex with respect to norm $\|\cdot\|$.

Substituting Eq.(41) into Eq. (40) and summing it over all iterations yields

$$
\begin{aligned}
\sum_{t=1}^{T} \sum_{k \in \mathcal{F}_t} [f_k(\boldsymbol{x}_{t,i_k}) - f_k(\boldsymbol{x}^*)] &\leq \sum_{t=1}^{T} \sum_{k \in \mathcal{F}_t} \langle \nabla f_k(\boldsymbol{x}_{t,i_k}), \boldsymbol{x}_{t,i_k} - \boldsymbol{x}^* \rangle \\
&\leq \frac{\eta}{2\sigma} T G_\star^2 + \frac{B_\psi(\boldsymbol{x}^*; \boldsymbol{x}_1)}{\eta} \\
&\quad + \sum_{t=1}^{T} \sum_{k \in \mathcal{F}_t} \left( \frac{4\xi R G_\star}{\eta} + G_\star \right) \|\boldsymbol{x}_{t,i_k+1} - \boldsymbol{y}^*_{t,i_k+1}\|.
\end{aligned}
\tag{42}
$$

The approximate solution $\boldsymbol{x}_{t,i_k+1}$ is given by

$$
\langle \nabla f_k(\boldsymbol{x}_{t,i_k}), \boldsymbol{x}_{t,i_k+1} \rangle + \frac{1}{\eta} B_\psi(\boldsymbol{x}_{t,i_k+1}; \boldsymbol{x}_{t,i_k}) \leq \left\langle \nabla f_k(\boldsymbol{x}_{t,i_k}), \boldsymbol{y}^*_{t,i_k+1} \right\rangle + \frac{1}{\eta} B_\psi(\boldsymbol{y}^*_{t,i_k+1}; \boldsymbol{x}_{t,i_k}) + \rho_{t,i_k}.
$$

Meanwhile, we define $A_{t,i_k}(\boldsymbol{x}) = \left\langle \nabla f_k(\boldsymbol{x}_{t,i_k}), \boldsymbol{x} \right\rangle + \frac{1}{\eta} B_\psi(\boldsymbol{x}; \boldsymbol{x}_{t,i_k})$ and $A_{t,i_k}(\boldsymbol{x})$ is $\frac{\sigma}{\eta}$-strongly convex with respect to norm $\|\cdot\|$, that is

$$
A_{t,i_k}(\boldsymbol{x}_{t,i_k+1}) - A_{t,i_k}(\boldsymbol{y}^*_{t,i_k+1}) \geq \langle \nabla A_{t,i_k}(\boldsymbol{y}^*_{t,i_k+1}), \boldsymbol{x}_{t,i_k+1} - \boldsymbol{y}^*_{t,i_k+1} \rangle + \frac{\sigma}{2\eta} \|\boldsymbol{x}_{t,i_k+1} - \boldsymbol{y}^*_{t,i_k+1}\|^2.
$$

Combining with the above, we obtain

$$
\|\boldsymbol{x}_{t,i_k+1} - \boldsymbol{y}^*_{t,i_k+1}\| \leq \sqrt{\frac{2\eta \rho_{t,i_k}}{\sigma}}.
\tag{43}
$$

By making $\rho_{t,i_k} = \frac{\eta^3}{2\sigma}$ and considering Eq. (42) together gives the result of Lemma 6

## D.2 Proof of Lemma 7

Due to the fact that $\psi(\boldsymbol{x})$ is $\sigma$-strongly convex with respect to norm $\|\cdot\|$. Thus

$$
B_\psi(\boldsymbol{y}^*_{t,i_k+1}; \boldsymbol{y}^*_{t,i_k}) + B_\psi(\boldsymbol{y}^*_{t,i_k}; \boldsymbol{y}^*_{t,i_k+1}) \geq \sigma \|\boldsymbol{y}^*_{t,i_k} - \boldsymbol{y}^*_{t,i_k+1}\|^2.
\tag{44}
$$

Meanwhile, based on the definition of Bregman divergence, the L.H.S of Eq. (44) can be rewritten as

$$
B_\psi(\boldsymbol{y}^*_{t,i_k+1}; \boldsymbol{y}^*_{t,i_k}) + B_\psi(\boldsymbol{y}^*_{t,i_k}; \boldsymbol{y}^*_{t,i_k+1}) = \left\langle \nabla \psi(\boldsymbol{y}^*_{t,i_k}) - \nabla \psi(\boldsymbol{y}^*_{t,i_k+1}), \boldsymbol{y}^*_{t,i_k} - \boldsymbol{y}^*_{t,i_k+1} \right\rangle.
$$

Recall back the optimality condition for the update rule, for any $\boldsymbol{y}^*_{t,i_k} \in \mathcal{X}$, we have

$$
\left\langle \eta \nabla f_k(\boldsymbol{x}_{t,i_k}) + \nabla \psi(\boldsymbol{y}^*_{t,i_k+1}) - \nabla \psi(\boldsymbol{x}_{t,i_k}), \boldsymbol{y}^*_{t,i_k} - \boldsymbol{y}^*_{t,i_k+1} \right\rangle \geq 0.
$$

Rearranging the terms of the above equality yields

$$
\begin{aligned}
&\left\langle \nabla \psi(\boldsymbol{y}^*_{t,i_k}) - \nabla \psi(\boldsymbol{y}^*_{t,i_k+1}), \boldsymbol{y}^*_{t,i_k} - \boldsymbol{y}^*_{t,i_k+1} \right\rangle \\
&\leq \langle \eta \nabla f_k(\boldsymbol{x}_{t,i_k}), \boldsymbol{y}^*_{t,i_k} - \boldsymbol{y}^*_{t,i_k+1} \rangle + \left\langle \nabla \psi(\boldsymbol{y}^*_{t,i_k}) - \nabla \psi(\boldsymbol{x}_{t,i_k}), \boldsymbol{y}^*_{t,i_k} - \boldsymbol{y}^*_{t,i_k+1} \right\rangle \\
&\leq (\eta G_\star + \xi G_\star \|\boldsymbol{x}_{t,i_k} - \boldsymbol{y}^*_{t,i_k}\|) \cdot \|\boldsymbol{y}^*_{t,i_k} - \boldsymbol{y}^*_{t,i_k+1}\|.
\end{aligned}
\tag{45}
$$

Considering Eq. (44) and Eq. (45), we obtain

$$
\|\boldsymbol{y}^*_{t,i_k} - \boldsymbol{y}^*_{t,i_k+1}\| \leq \frac{\eta G_\star + \xi G_\star \|\boldsymbol{x}_{t,i_k} - \boldsymbol{y}^*_{t,i_k}\|}{\sigma}.
$$

Combining with Eq. (43) gives the the result of Lemma 7.

$$
\begin{aligned}
\|\boldsymbol{x}_{t,i_k+1} - \boldsymbol{x}_{t,i_k}\| &\leq \|\boldsymbol{y}^*_{t,i_k+1} - \boldsymbol{y}^*_{t,i_k}\| + \|\boldsymbol{x}_{t,i_k} - \boldsymbol{y}^*_{t,i_k}\| + \|\boldsymbol{x}_{t,i_k+1} - \boldsymbol{y}^*_{t,i_k+1}\| \\
&\leq \frac{\eta G_\star + 2\eta^2}{\sigma} + \frac{\eta^2 \xi G_\star}{\sigma^2}.
\end{aligned}
$$

# E PROOF OF THEOREM 4

## E.1 Proof of Lemma 8

Due to the relative strong convexity, for the normal term of Eq. (18), we have

$$\sum_{t=1}^{T} \sum_{k \in \mathcal{F}_t} [f_k(x_t) - f_k(x^*)] \leq \sum_{t=1}^{T} \sum_{k \in \mathcal{F}_t} [\langle \nabla f_k(x_t), x_t - x^* \rangle - \gamma B_\psi(x^*; x_t)]$$

$$= \sum_{t=1}^{T} \sum_{k \in \mathcal{F}_t} \langle \nabla f_k(x_t), x_t - x^* \rangle - \sum_{t=1}^{T} |\mathcal{F}_t| \gamma B_\psi(x^*; x_t). \tag{46}$$

At each iteration $t$, the decision $y_{t+1}^*$ is updated by

$$y_{t+1}^* = \arg\min_{x \in \mathcal{X}} \left\{ \sum_{k \in \mathcal{F}_t} \langle \nabla f_k(x_t), x \rangle + \frac{1}{\eta_t} B_\psi(x; x_t) \right\}.$$

Due to the optimality condition of the update rule, for any $x^* \in \mathcal{X}$, we have

$$\left\langle \eta_t \sum_{k \in \mathcal{F}_t} \nabla f_k(x_t) + \nabla \psi(y_{t+1}^*) - \nabla \psi(x_t), x^* - y_{t+1}^* \right\rangle \geq 0. \tag{47}$$

Rearranging the terms of Eq. (47) gives

$$\sum_{k \in \mathcal{F}_t} \langle \nabla f_k(x_t), x_t - x^* \rangle \leq \sum_{k \in \mathcal{F}_t} \langle \nabla f_k(x_t), x_t - y_{t+1}^* \rangle + \frac{\langle \nabla \psi(y_{t+1}^*) - \nabla \psi(x_t), x^* - y_{t+1}^* \rangle}{\eta_t}$$

$$\leq \sum_{k \in \mathcal{F}_t} \langle \nabla f_k(x_t), x_t - x_{t+1} \rangle + \sum_{k \in \mathcal{F}_t} \langle \nabla f_k(x_t), x_{t+1} - y_{t+1}^* \rangle$$

$$+ \frac{\langle \nabla \psi(y_{t+1}^*) - \nabla \psi(x_t), x^* - y_{t+1}^* \rangle}{\eta_t}.$$

For the last term above, we have

$$\langle \nabla \psi(y_{t+1}^*) - \nabla \psi(x_t), x^* - y_{t+1}^* \rangle = \langle \nabla \psi(y_{t+1}^*) - \nabla \psi(x_{t+1}), x^* - y_{t+1}^* \rangle$$

$$+ \langle \nabla \psi(x_{t+1}) - \nabla \psi(x_t), x^* - x_{t+1} \rangle$$

$$+ \langle \nabla \psi(x_{t+1}) - \nabla \psi(x_t), x_{t+1} - y_{t+1}^* \rangle$$

$$\leq \langle \nabla \psi(x_{t+1}) - \nabla \psi(x_t), x^* - x_{t+1} \rangle$$

$$+ 4\xi RG_\star \|x_{t+1} - y_{t+1}^*\|.$$

The last inequality is due to $\psi$ has $\xi G_\star$-Lipschitz gradients.

Applying the three-point identity of Bregman divergence to the above formula gives

$$\sum_{k \in \mathcal{F}_t} \langle \nabla f_k(x_t), x_t - x^* \rangle$$

$$\leq \sum_{k \in \mathcal{F}_t} \langle \nabla f_k(x_t), x_t - x_{t+1} \rangle + \frac{B_\psi(x^*; x_t) - B_\psi(x^*; x_{t+1}) - B_\psi(x_{t+1}; x_t)}{\eta_t}$$

$$+ \left( |\mathcal{F}_t| G_\star + \frac{4\xi RG_\star}{\eta_t} \right) \cdot \|x_{t+1} - y_{t+1}^*\|$$

$$\leq \frac{\eta_t |\mathcal{F}_t|^2 G_\star^2}{2\sigma} + \frac{\sigma}{2\eta_t} \|x_t - x_{t+1}\|^2 + \frac{B_\psi(x^*; x_t) - B_\psi(x^*; x_{t+1}) - B_\psi(x_{t+1}; x_t)}{\eta_t}$$

$$+ \left( |\mathcal{F}_t| G_\star + \frac{4\xi RG_\star}{\eta_t} \right) \cdot \|x_{t+1} - y_{t+1}^*\|$$

$$\leq \frac{\eta_t |\mathcal{F}_t|^2 G_\star^2}{2\sigma} + \frac{B_\psi(x^*; x_t) - B_\psi(x^*; x_{t+1})}{\eta_t} + \left( |\mathcal{F}_t| G_\star + \frac{4\xi RG_\star}{\eta_t} \right) \cdot \|x_{t+1} - y_{t+1}^*\|. \tag{48}$$

The second inequality is due to $\sum_{k \in \mathcal{F}_t} \langle \nabla f_k(x_t), x_t - x_{t+1} \rangle \leq |\mathcal{F}_t| G_\star \|x_t - x_{t+1}\| \leq \frac{\eta_t |\mathcal{F}_t|^2 G_\star^2}{2\sigma} + \frac{\sigma}{2\eta_t} \|x_t - x_{t+1}\|^2$. The last inequality is due to $B_\psi(x_{t+1}; x_t) \geq \frac{\sigma}{2} \|x_t - x_{t+1}\|^2$.

Now we discuss the error incurred by the approximate solution, given by

$$\sum_{k \in \mathcal{F}_t} \langle \nabla f_k(\boldsymbol{x}_t), \boldsymbol{x}_{t+1} \rangle + \frac{1}{\eta_t} B_\psi(\boldsymbol{x}_{t+1}; \boldsymbol{x}_t) \le \sum_{k \in \mathcal{F}_t} \langle \nabla f_k(\boldsymbol{x}_t), \boldsymbol{y}_{t+1}^* \rangle + \frac{1}{\eta_t} B_\psi(\boldsymbol{y}_{t+1}^*; \boldsymbol{x}_t) + \rho_t.$$

Meanwhile, we make $B_t(\boldsymbol{x}) = \sum_{k \in \mathcal{F}_t} \langle \nabla f_k(\boldsymbol{x}_t), \boldsymbol{x} \rangle + \frac{1}{\eta_t} B_\psi(\boldsymbol{x}; \boldsymbol{x}_t)$ and note that $B_t$ is $\frac{\sigma}{\eta_t}$-strongly with respect to norm $\|\cdot\|$, that is

$$B_t(\boldsymbol{x}_{t+1}) - B_t(\boldsymbol{y}_{t+1}^*) \ge \langle \nabla B_t(\boldsymbol{y}_{t+1}^*), \boldsymbol{x}_{t+1} - \boldsymbol{y}_{t+1}^* \rangle + \frac{\sigma}{2\eta_t} \|\boldsymbol{x}_{t+1} - \boldsymbol{y}_{t+1}^*\|^2$$

$$\ge \frac{\sigma}{2\eta_t} \|\boldsymbol{x}_{t+1} - \boldsymbol{y}_{t+1}^*\|^2.$$

The last inequality is due to the optimality condition.

By combining above results with $\rho_t = \frac{\eta_t^3}{2\sigma}$, we obtain

$$\|\boldsymbol{x}_{t+1} - \boldsymbol{y}_{t+1}^*\| \le \sqrt{\frac{2\eta_t \rho_t}{\sigma}} = \frac{\eta_t^2}{\sigma}.$$

Note that $\eta_t = \frac{1}{\sum_{\tau=1}^{t} |\mathcal{F}_\tau| \gamma}$, summing Eq. (48) over all iterations and substituting it into Eq. (46), we have

$$\sum_{t=1}^{T} \sum_{k \in \mathcal{F}_t} [f_k(\boldsymbol{x}_t) - f_k(\boldsymbol{x}^*)]$$

$$\le \sum_{t=1}^{T} \frac{\eta_t |\mathcal{F}_t|^2 G_\star^2}{2\sigma} + \sum_{t=1}^{T} \left[ \left( \frac{1}{\eta_t} - |\mathcal{F}_t|\gamma \right) B_\psi(\boldsymbol{x}^*; \boldsymbol{x}_t) - \frac{1}{\eta_t} B_\psi(\boldsymbol{x}^*; \boldsymbol{x}_{t+1}) \right] + \frac{4\xi R G_\star (1 + \ln T)}{\sigma \gamma} + \frac{2d G_\star}{\sigma \gamma^2} \tag{49}$$

$$\le \sum_{t=1}^{T} \frac{d |\mathcal{F}_t| G_\star^2}{2\sigma \gamma \sum_{\tau=1}^{t} |\mathcal{F}_\tau|} + \sum_{t=1}^{T} \left[ \sum_{\tau=1}^{t-1} |\mathcal{F}_\tau| \gamma B_\psi(\boldsymbol{x}^*; \boldsymbol{x}_t) - \sum_{\tau=1}^{t} |\mathcal{F}_\tau| \gamma B_\psi(\boldsymbol{x}^*; \boldsymbol{x}_{t+1}) \right] + \frac{4\xi R G_\star (1 + \ln T)}{\sigma \gamma} + \frac{2d G_\star}{\sigma \gamma^2}$$

$$\le \frac{(d G_\star^2 + 8\xi R G_\star)(1 + \ln T)}{2\sigma \gamma} + \frac{2d G_\star}{\sigma \gamma^2}.$$

## E.2 Proof of Lemma 9

Due to the fact that $\psi(\cdot)$ is $\sigma$-strongly convex with respect to norm $\|\cdot\|$, we have

$$B_\psi(\boldsymbol{y}_{t+1}^*; \boldsymbol{y}_t^*) + B_\psi(\boldsymbol{y}_t^*; \boldsymbol{y}_{t+1}^*) \ge \sigma \|\boldsymbol{y}_t^* - \boldsymbol{y}_{t+1}^*\|^2. \tag{50}$$

Meanwhile, the upper bound of the L.H.S of Eq. (50) can be rewritten as

$$B_\psi(\boldsymbol{y}_{t+1}^*; \boldsymbol{y}_t^*) + B_\psi(\boldsymbol{y}_t^*; \boldsymbol{y}_{t+1}^*) = \langle \nabla \psi(\boldsymbol{y}_t^*) - \nabla \psi(\boldsymbol{y}_{t+1}^*), \boldsymbol{y}_t^* - \boldsymbol{y}_{t+1}^* \rangle.$$

Due to the optimality condition of the update rule, for any $\boldsymbol{y}_t^* \in \mathcal{X}$, we have

$$\left\langle \eta_t \sum_{k \in \mathcal{F}_t} \nabla f_k(\boldsymbol{x}_t) + \nabla \psi(\boldsymbol{y}_{t+1}^*) - \nabla \psi(\boldsymbol{x}_t), \boldsymbol{y}_t^* - \boldsymbol{y}_{t+1}^* \right\rangle \ge 0.$$

Rearranging the terms gives

$$\langle \nabla \psi(\boldsymbol{y}_t^*) - \nabla \psi(\boldsymbol{y}_{t+1}^*), \boldsymbol{y}_t^* - \boldsymbol{y}_{t+1}^* \rangle$$

$$\le \eta_t \sum_{k \in \mathcal{F}_t} \langle \nabla f_k(\boldsymbol{x}_t), \boldsymbol{y}_t^* - \boldsymbol{y}_{t+1}^* \rangle + \langle \nabla \psi(\boldsymbol{y}_t^*) - \nabla \psi(\boldsymbol{x}_t), \boldsymbol{y}_t^* - \boldsymbol{y}_{t+1}^* \rangle \tag{51}$$

$$\le (\eta_t |\mathcal{F}_t| G_\star + \xi G_\star \|\boldsymbol{x}_t - \boldsymbol{y}_t^*\|) \cdot \|\boldsymbol{y}_t^* - \boldsymbol{y}_{t+1}^*\|.$$

Combining with Eq. (50) and Eq. (51) gives

$$\|\boldsymbol{y}_{t+1}^* - \boldsymbol{y}_t^*\| \le \frac{\eta_t |\mathcal{F}_t| G_\star + \xi G_\star \|\boldsymbol{x}_t - \boldsymbol{y}_t^*\|}{\sigma}.$$

Then we get the result of Lemma 9 as follows.

$$\|\boldsymbol{x}_{t+1} - \boldsymbol{x}_t\| \le \|\boldsymbol{y}_{t+1}^* - \boldsymbol{y}_t^*\| + \|\boldsymbol{x}_{t+1} - \boldsymbol{y}_{t+1}^*\| + \|\boldsymbol{x}_t - \boldsymbol{y}_t^*\|$$

$$\le \frac{\eta_t |\mathcal{F}_t| G_\star + \eta_t^2 + \eta_{t-1}^2}{\sigma} + \frac{\eta_{t-1}^2 \xi G_\star}{\sigma^2}.$$

# F PROOF OF THEOREM 5

When the received feedback is the value of the loss function's gradient in the corresponding decision point. In the case of general convexity, the regret format is

$$
\begin{aligned}
\text{Reg}_T &= \sum_{t=1}^{T} [f_t(\boldsymbol{x}_t) - f_t(\boldsymbol{x}^*)] \\
&= \sum_{t=1}^{T} \sum_{k \in \mathcal{F}_t} [f_k(\boldsymbol{x}_k) - f_k(\boldsymbol{x}^*)] \\
&\leq \sum_{t=1}^{T} \sum_{k \in \mathcal{F}_t} \langle \nabla f_k(\boldsymbol{x}_k), \boldsymbol{x}_k - \boldsymbol{x}^* \rangle \\
&= \underbrace{\sum_{t=1}^{T} \sum_{k \in \mathcal{F}_t} \langle \nabla f_k(\boldsymbol{x}_k), \boldsymbol{x}_{t,i_k} - \boldsymbol{x}^* \rangle}_{\text{normal term}} + \underbrace{\sum_{t=1}^{T} \sum_{k \in \mathcal{F}_t} \langle \nabla f_k(\boldsymbol{x}_k), \boldsymbol{x}_k - \boldsymbol{x}_{t,i_k} \rangle}_{\text{delayed term}}.
\end{aligned}
\tag{52}
$$

We first analyze the normal term of Eq. (52).

LEMMA 10. *The normal term of Eq. (52) is bounded by*

$$
\sum_{t=1}^{T} \sum_{k \in \mathcal{F}_t} \langle \nabla f_k(\boldsymbol{x}_k), \boldsymbol{x}_{t,i_k} - \boldsymbol{x}^* \rangle \leq \frac{\eta T(G_\star^2 + 8\xi R G_\star + 2\eta G_\star)}{2\sigma} + \frac{B_\psi(\boldsymbol{x}^*; \boldsymbol{x}_1)}{\eta}.
\tag{53}
$$

Next, we discuss the delayed term of Eq. (52).

$$
\begin{aligned}
&\sum_{t=1}^{T} \sum_{k \in \mathcal{F}_t} \langle \nabla f_k(\boldsymbol{x}_k), \boldsymbol{x}_k - \boldsymbol{x}_{t,i_k} \rangle \\
&\leq \sum_{t=1}^{T} \sum_{k \in \mathcal{F}_t} G_\star \|\boldsymbol{x}_{t,i_k} - \boldsymbol{x}_k\| \\
&\leq \sum_{t=1}^{T} \sum_{k \in \mathcal{F}_t} G_\star \left( \sum_{\tau=k}^{t-1} \sum_{s \in \mathcal{F}_\tau} \|\boldsymbol{x}_{\tau,i_s+1} - \boldsymbol{x}_{\tau,i_s}\| + \sum_{s \in \mathcal{F}_{t,k}} \|\boldsymbol{x}_{t,i_s+1} - \boldsymbol{x}_{t,i_s}\| \right).
\end{aligned}
\tag{54}
$$

From the above formula, the crucial key impacting the bound of the delayed term is the gap between $\boldsymbol{x}_{t,i_s}$ and $\boldsymbol{x}_{t,i_s+1}$.

LEMMA 11. *For each $t \in [T], k \in \mathcal{F}_t$, our SDMD-GC algorithm ensures that*

$$
\|\boldsymbol{x}_{t,i_k+1} - \boldsymbol{x}_{t,i_k}\| \leq \frac{\eta G_\star + 2\eta^2}{\sigma} + \frac{\eta^2 \xi G_\star}{\sigma^2}.
\tag{55}
$$

Substituting Eq. (55) into Eq. (54) gives

$$
\sum_{t=1}^{T} \sum_{k \in \mathcal{F}_t} \langle \nabla f_k(\boldsymbol{x}_k), \boldsymbol{x}_k - \boldsymbol{x}_{t,i_k} \rangle \leq 2D_T G_\star \left( \frac{\eta G_\star + 2\eta^2}{\sigma} + \frac{\eta^2 \xi G_\star}{\sigma^2} \right).
\tag{56}
$$

Combining with Eq. (53) and Eq. (56), we get the result of Theorem 5.

## F.1 Proof of Lemma 10

Note that at each iteration $t$, the decision $\boldsymbol{y}_{t,i_k+1}^*$ is updated by

$$
\boldsymbol{y}_{t,i_k+1}^* = \arg\min_{\boldsymbol{x} \in \mathcal{X}} \left\{ \langle \nabla f_k(\boldsymbol{x}_k), \boldsymbol{x} \rangle + \frac{1}{\eta} B_\psi(\boldsymbol{x}; \boldsymbol{x}_{t,i_k}) \right\}.
$$

From the optimality condition for the update of SDMD-GC, for any $\boldsymbol{x}^* \in \mathcal{X}$, we have

$$
\left\langle \eta \nabla f_k(\boldsymbol{x}_k) + \nabla \psi(\boldsymbol{y}_{t,i_k+1}^*) - \nabla \psi(\boldsymbol{x}_{t,i_k}), \boldsymbol{x}^* - \boldsymbol{y}_{t,i_k+1}^* \right\rangle \geq 0.
$$

Rearranging the terms above yields

$$
\begin{aligned}
\langle \nabla f_k(x_k), x_{t,i_k} - x^* \rangle & \\
\leq & \langle \nabla f_k(x_k), x_{t,i_k} - y^*_{t,i_k+1} \rangle + \frac{\langle \nabla \psi(y^*_{t,i_k+1}) - \nabla \psi(x_{t,i_k}), x^* - y^*_{t,i_k+1} \rangle}{\eta} \\
\leq & \langle \nabla f_k(x_k), x_{t,i_k} - x_{t,i_k+1} \rangle + G_\star \| x_{t,i_k+1} - y^*_{t,i_k+1} \| \\
& + \frac{\langle \nabla \psi(y^*_{t,i_k+1}) - \nabla \psi(x_{t,i_k}), x^* - y^*_{t,i_k+1} \rangle}{\eta}.
\end{aligned}
\tag{57}
$$

For the last term above, we have

$$
\begin{aligned}
\langle \nabla \psi(y^*_{t,i_k+1}) - \nabla \psi(x_{t,i_k}), x^* - y^*_{t,i_k+1} \rangle = & \langle \nabla \psi(x_{t,i_k+1}) - \nabla \psi(x_{t,i_k}), x^* - x_{t,i_k+1} \rangle \\
& + \langle \nabla \psi(y^*_{t,i_k+1}) - \nabla \psi(x_{t,i_k+1}), x^* - y^*_{t,i_k+1} \rangle \\
& + \langle \nabla \psi(x_{t,i_k+1}) - \nabla \psi(x_{t,i_k}), x_{t,i_k+1} - y^*_{t,i_k+1} \rangle \\
\leq & \langle \nabla \psi(x_{t,i_k+1}) - \nabla \psi(x_{t,i_k}), x^* - x_{t,i_k+1} \rangle \\
& + 4\xi R G_\star \| x_{t,i_k+1} - y^*_{t,i_k+1} \|.
\end{aligned}
$$

The three-point identity regarding Bregman divergence is

$$
\langle \nabla \psi(x_{t,i_k+1}) - \nabla \psi(x_{t,i_k}), x^* - x_{t,i_k+1} \rangle = B_\psi(x^*; x_{t,i_k}) - B_\psi(x^*; x_{t,i_k+1}) - B_\psi(x_{t,i_k+1}; x_{t,i_k}).
$$

Applying the three-point identity to Eq.(57) gives

$$
\begin{aligned}
\langle \nabla f_k(x_k), x_{t,i_k} - x^* \rangle \leq & \langle \nabla f_k(x_k), x_{t,i_k} - x_{t,i_k+1} \rangle + G_\star \| x_{t,i_k+1} - y^*_{t,i_k+1} \| \\
& + \frac{B_\psi(x^*; x_{t,i_k}) - B_\psi(x^*; x_{t,i_k+1}) - B_\psi(x_{t,i_k+1}; x_{t,i_k})}{\eta} \\
& + \frac{4\xi R G_\star \| x_{t,i_k+1} - y^*_{t,i_k+1} \|}{\eta}.
\end{aligned}
\tag{58}
$$

For the first term on the R.H.S of Eq.(58), we have

$$
\begin{aligned}
\langle \nabla f_k(x_k), x_{t,i_k} - x_{t,i_k+1} \rangle \leq & G_\star \| x_{t,i_k} - x_{t,i_k+1} \| \\
\leq & \frac{\eta}{2\sigma} G_\star^2 + \frac{\sigma}{2\eta} \| x_{t,i_k} - x_{t,i_k+1} \|^2.
\end{aligned}
\tag{59}
$$

Additionally, we assume the regularization function $\psi(\cdot)$ is $\sigma$-strongly convex with respect to norm $\| \cdot \|$. Then we get

$$
B_\psi(x_{t,i_k+1}; x_{t,i_k}) \geq \frac{\sigma}{2} \| x_{t,i_k} - x_{t,i_k+1} \|^2.
\tag{60}
$$

Substituting Eq.(59) and Eq.(60) into Eq. (58) and summing it over all iterations yields

$$
\begin{aligned}
\sum_{t=1}^{T} \sum_{k \in \mathcal{F}_t} \langle \nabla f_k(x_k), x_{t,i_k} - x^* \rangle \leq & \sum_{t=1}^{T} \sum_{k \in \mathcal{F}_t} \frac{\eta}{2\sigma} G_\star^2 + \frac{B_\psi(x^*; x_1) - B_\psi(x^*; x_{T+1})}{\eta} \\
& + \sum_{t=1}^{T} \sum_{k \in \mathcal{F}_t} \left( \frac{4\xi R G_\star}{\eta} + G_\star \right) \| x_{t,i_k+1} - y^*_{t,i_k+1} \| \\
\leq & \frac{\eta}{2\sigma} T G_\star^2 + \frac{B_\psi(x^*; x_1)}{\eta} \\
& + \sum_{t=1}^{T} \sum_{k \in \mathcal{F}_t} \left( \frac{4\xi R G_\star}{\eta} + G_\star \right) \| x_{t,i_k+1} - y^*_{t,i_k+1} \|.
\end{aligned}
\tag{61}
$$

The last inequality above is due to $B_\psi(x^*; x_{T+1}) \geq 0$ for any convex regularization function $\psi$.

Call back the approximate solution of $x_{t,i_k+1}$, that is

$$
\langle \nabla f_k(x_k), x_{t,i_k+1} \rangle + \frac{1}{\eta} B_\psi(x_{t,i_k+1}; x_{t,i_k}) \leq \langle \nabla f_k(x_k), y^*_{t,i_k+1} \rangle + \frac{1}{\eta} B_\psi(y^*_{t,i_k+1}; x_{t,i_k}) + \rho_{t,i_k}.
$$

We make $C_{t,i_k}(x) = \langle \nabla f_k(x_k), x \rangle + \frac{1}{\eta} B_\psi(x; x_{t,i_k})$. Utilizing its strong convexity, we have

$$C_{t,i_k}(x_{t,i_k+1}) - C_{t,i_k}(y^*_{t,i_k+1}) \geq \langle \nabla C_{t,i_k}(y^*_{t,i_k+1}), x_{t,i_k+1} - y^*_{t,i_k+1} \rangle + \frac{\sigma}{2\eta} \|x_{t,i_k+1} - y^*_{t,i_k+1}\|^2$$

$$\geq \frac{\sigma}{2\eta} \|x_{t,i_k+1} - y^*_{t,i_k+1}\|^2.$$

Considering the above together, we obtain

$$\|x_{t,i_k+1} - y^*_{t,i_k+1}\| \leq \sqrt{\frac{2\eta \rho_{t,i_k}}{\sigma}}. \tag{62}$$

Substituting Eq. (62) into Eq. (61) and making $\rho_{t,i_k} = \frac{\eta^3}{2\sigma}$ gives the result of Lemma 10.

## F.2 Proof of Lemma 11

Due to the fact that $\psi(\cdot)$ is $\sigma$-strongly convex with respect to norm $\|\cdot\|$. Thus

$$B_\psi(y^*_{t,i_k+1}; y^*_{t,i_k}) + B_\psi(y^*_{t,i_k}; y^*_{t,i_k+1}) \geq \sigma \|y^*_{t,i_k} - y^*_{t,i_k+1}\|^2. \tag{63}$$

Meanwhile, based on the definition of Bregman divergence, the L.H.S of Eq. (63) can be rewritten as

$$B_\psi(y^*_{t,i_k+1}; y^*_{t,i_k}) + B_\psi(y^*_{t,i_k}; y^*_{t,i_k+1}) = \left\langle \nabla \psi(y^*_{t,i_k}) - \nabla \psi(y^*_{t,i_k+1}), y^*_{t,i_k} - y^*_{t,i_k+1} \right\rangle.$$

Recall back the optimality condition for the update rule, for any $y^*_{t,i_k} \in \mathcal{X}$ we have

$$\left\langle \eta \nabla f_k(x_k) + \nabla \psi(y^*_{t,i_k+1}) - \nabla \psi(x_{t,i_k}), y^*_{t,i_k} - y^*_{t,i_k+1} \right\rangle \geq 0.$$

Rearranging the terms of the above equality yields

$$\left\langle \nabla \psi(y^*_{t,i_k}) - \nabla \psi(y^*_{t,i_k+1}), y^*_{t,i_k} - y^*_{t,i_k+1} \right\rangle \leq \langle \eta \nabla f_k(x_k), y^*_{t,i_k} - y^*_{t,i_k+1} \rangle$$

$$+ \left\langle \nabla \psi(y^*_{t,i_k}) - \nabla \psi(x_{t,i_k}), y^*_{t,i_k} - y^*_{t,i_k+1} \right\rangle$$

$$\leq \eta G_\star \|y^*_{t,i_k} - y^*_{t,i_k+1}\|$$

$$+ \xi G_\star \|x_{t,i_k} - y^*_{t,i_k}\| \cdot \|y^*_{t,i_k} - y^*_{t,i_k+1}\|. \tag{64}$$

Considering Eq. (63) and Eq. (64), we obtain

$$\|y^*_{t,i_k} - y^*_{t,i_k+1}\| \leq \frac{\eta G_\star + \xi G_\star \|x_{t,i_k} - y^*_{t,i_k}\|}{\sigma}.$$

Combining with Eq. (62) gives

$$\|x_{t,i_k+1} - x_{t,i_k}\| \leq \|y^*_{t,i_k+1} - y^*_{t,i_k}\| + \|x_{t,i_k+1} - y^*_{t,i_k+1}\| + \|x_{t,i_k} - y^*_{t,i_k}\|$$

$$\leq \frac{\eta G_\star + 2\eta^2}{\sigma} + \frac{\eta^2 \xi G_\star}{\sigma^2}.$$

## G PROOF OF THEOREM 6

In the case of relative strong convexity, the regret format is

$$\text{Reg}_T = \sum_{t=1}^{T} [f_t(x_t) - f_t(x^*)]$$

$$\leq \sum_{t=1}^{T} \sum_{k \in \mathcal{F}_t} \langle \nabla f_k(x_k), x_k - x^* \rangle - \sum_{t=1}^{T} \gamma B_\psi(x^*; x_t) \tag{65}$$

$$= \underbrace{\sum_{t=1}^{T} \sum_{k \in \mathcal{F}_t} \langle \nabla f_k(x_k), x_t - x^* \rangle - \sum_{t=1}^{T} \gamma B_\psi(x^*; x_t)}_{\text{normal term}} + \underbrace{\sum_{t=1}^{T} \sum_{k \in \mathcal{F}_t} \langle \nabla f_k(x_k), x_k - x_t \rangle}_{\text{delayed term}}.$$

The bound of the normal term in Eq. (65) is as follows.

Lemma 12. *The normal term of Eq. (65) is bounded by*

$$\sum_{t=1}^{T} \sum_{k \in \mathcal{F}_t} \langle \nabla f_k(x_k), x_t - x^* \rangle - \sum_{t=1}^{T} \gamma B_\psi(x^*; x_t) \leq \frac{2dG_\star}{\sigma\gamma^2} + \frac{(dG_\star^2 + 8\xi R G_\star)(1 + \ln T)}{2\sigma\gamma}. \tag{66}$$

Next, we discuss the delayed term of Eq. (65).

$$\sum_{t=1}^{T}\sum_{k\in\mathcal{F}_t}\langle\nabla f_k(\boldsymbol{x}_k),\boldsymbol{x}_k-\boldsymbol{x}_t\rangle \leq \sum_{t=1}^{T}\sum_{k\in\mathcal{F}_t}\|\nabla f_k(\boldsymbol{x}_k)\|_{\star}\cdot\|\boldsymbol{x}_k-\boldsymbol{x}_t\|$$

$$\leq \sum_{t=1}^{T}\sum_{k\in\mathcal{F}_t}G_{\star}\sum_{\tau=k}^{k+d_k-1}\|\boldsymbol{x}_{\tau+1}-\boldsymbol{x}_{\tau}\| \tag{67}$$

$$\leq dG_{\star}\sum_{t=1}^{T}\|\boldsymbol{x}_{t+1}-\boldsymbol{x}_t\|.$$

The key is the difference between $\boldsymbol{x}_t$ and $\boldsymbol{x}_{t+1}$.

LEMMA 13. *For each $t\in[T]$, our SDMD-RSC algorithm ensures that*

$$\|\boldsymbol{x}_{t+1}-\boldsymbol{x}_t\| \leq \frac{\eta_t|\mathcal{F}_t|G_{\star}+\eta_t^2+\eta_{t-1}^2}{\sigma}+\frac{\eta_{t-1}^2\xi G_{\star}}{\sigma^2}. \tag{68}$$

Note that we make $\eta_0=\eta_1$. Considering $\eta_t=\frac{1}{\gamma t}$ and substituting Eq. (68) into Eq. (67) gives

$$\sum_{t=1}^{T}\sum_{k\in\mathcal{F}_t}\langle\nabla f_k(\boldsymbol{x}_k),\boldsymbol{x}_k-\boldsymbol{x}_t\rangle \leq \frac{dG_{\star}^2(1+\ln T)}{\sigma\gamma}+\frac{4dG_{\star}}{\sigma\gamma^2}+\frac{2d\xi G_{\star}}{\sigma^2\gamma^2}. \tag{69}$$

Combining with Eq. (66) and Eq. (69), we get the result of Theorem 6.

## G.1 Proof of Lemma 12

At each iteration $t$, the decision $\boldsymbol{y}_{t+1}^{*}$ is updated by

$$\boldsymbol{y}_{t+1}^{*}=\arg\min_{\boldsymbol{x}\in\mathcal{X}}\left\{\sum_{k\in\mathcal{F}_t}\langle\nabla f_k(\boldsymbol{x}_k),\boldsymbol{x}\rangle+\frac{1}{\eta_t}B_{\psi}(\boldsymbol{x};\boldsymbol{x}_t)\right\}.$$

From the optimality condition for the update rule, for any $\boldsymbol{x}^{*}\in\mathcal{X}$, we have

$$\left\langle\eta_t\sum_{k\in\mathcal{F}_t}\nabla f_k(\boldsymbol{x}_k)+\nabla\psi(\boldsymbol{y}_{t+1}^{*})-\nabla\psi(\boldsymbol{x}_t),\boldsymbol{x}^{*}-\boldsymbol{y}_{t+1}^{*}\right\rangle\geq 0.$$

Rearranging the terms above yields

$$\sum_{k\in\mathcal{F}_t}\left\langle\nabla f_k(\boldsymbol{x}_k),\boldsymbol{x}_t-\boldsymbol{x}^{*}\right\rangle \leq \sum_{k\in\mathcal{F}_t}\left\langle\nabla f_k(\boldsymbol{x}_k),\boldsymbol{x}_t-\boldsymbol{y}_{t+1}^{*}\right\rangle+\frac{\left\langle\nabla\psi(\boldsymbol{y}_{t+1}^{*})-\nabla\psi(\boldsymbol{x}_t),\boldsymbol{x}^{*}-\boldsymbol{y}_{t+1}^{*}\right\rangle}{\eta_t}$$

$$\leq \sum_{k\in\mathcal{F}_t}\left\langle\nabla f_k(\boldsymbol{x}_k),\boldsymbol{x}_t-\boldsymbol{x}_{t+1}\right\rangle+|\mathcal{F}_t|G_{\star}\cdot\|\boldsymbol{x}_{t+1}-\boldsymbol{y}_{t+1}^{*}\|$$

$$+\frac{\left\langle\nabla\psi(\boldsymbol{y}_{t+1}^{*})-\nabla\psi(\boldsymbol{x}_t),\boldsymbol{x}^{*}-\boldsymbol{y}_{t+1}^{*}\right\rangle}{\eta_t}.$$

For the last term above, we have

$$\left\langle\nabla\psi(\boldsymbol{y}_{t+1}^{*})-\nabla\psi(\boldsymbol{x}_t),\boldsymbol{x}^{*}-\boldsymbol{y}_{t+1}^{*}\right\rangle\leq\left\langle\nabla\psi(\boldsymbol{y}_{t+1}^{*})-\nabla\psi(\boldsymbol{x}_{t+1}),\boldsymbol{x}^{*}-\boldsymbol{y}_{t+1}^{*}\right\rangle$$

$$+\left\langle\nabla\psi(\boldsymbol{x}_{t+1})-\nabla\psi(\boldsymbol{x}_t),\boldsymbol{x}^{*}-\boldsymbol{x}_{t+1}\right\rangle$$

$$+\left\langle\nabla\psi(\boldsymbol{x}_{t+1})-\nabla\psi(\boldsymbol{x}_t),\boldsymbol{x}_{t+1}-\boldsymbol{y}_{t+1}^{*}\right\rangle$$

$$\leq\left\langle\nabla\psi(\boldsymbol{x}_{t+1})-\nabla\psi(\boldsymbol{x}_t),\boldsymbol{x}^{*}-\boldsymbol{x}_{t+1}\right\rangle$$

$$+4\xi RG_{\star}\|\boldsymbol{x}_{t+1}-\boldsymbol{y}_{t+1}^{*}\|.$$

Applying the three-point identity of Bregman divergence gives

$$\left\langle\nabla\psi(\boldsymbol{x}_{t+1})-\nabla\psi(\boldsymbol{x}_t),\boldsymbol{x}_{t+1}-\boldsymbol{y}_{t+1}^{*}\right\rangle\leq B_{\psi}(\boldsymbol{x}^{*};\boldsymbol{x}_t)-B_{\psi}(\boldsymbol{x}^{*};\boldsymbol{x}_{t+1})-B_{\psi}(\boldsymbol{x}_{t+1};\boldsymbol{x}_t). \tag{70}$$

For the first term on the R.H.S of Eq. (70), we have

$$\sum_{k\in\mathcal{F}_t}\langle\nabla f_k(\boldsymbol{x}_k),\boldsymbol{x}_t-\boldsymbol{x}_{t+1}\rangle\leq|\mathcal{F}_t|G_{\star}\|\boldsymbol{x}_t-\boldsymbol{x}_{t+1}\|\leq\frac{\eta_t}{2\sigma}|\mathcal{F}_t|^2G_{\star}^2+\frac{\sigma}{2\eta_t}\|\boldsymbol{x}_t-\boldsymbol{x}_{t+1}\|^2. \tag{71}$$

Additionally, we assume the regularization function $\psi(\cdot)$ is $\sigma$-strongly convex with respect to norm $\|\cdot\|$. Then we have

$$B_\psi(x_{t+1}; x_t) \geq \frac{\sigma}{2}\|x_t - x_{t+1}\|^2. \tag{72}$$

Combining with Eq. (70), Eq. (71) and Eq. (72) yields

$$\sum_{t=1}^{T}\sum_{k\in\mathcal{F}_t}\langle\nabla f_k(x_k), x_t - x^*\rangle \leq \sum_{t=1}^{T}\frac{\eta_t}{2\sigma}|\mathcal{F}_t|^2 G_\star^2 + \sum_{t=1}^{T}\frac{B_\psi(x^*; x_t) - B_\psi(x^*; x_{t+1})}{\eta_t}$$
$$+ \sum_{t=1}^{T}\left(|\mathcal{F}_t|G_\star + \frac{4\xi RG_\psi}{\eta_t}\right)\|x_{t+1} - y_{t+1}^*\|. \tag{73}$$

Subtracting $\sum_{t=1}^{T}\gamma B_\psi(x^*; x_t)$ from Eq. (73) and substituting $\eta_t = \frac{1}{\gamma t}$, we obtain

$$\sum_{t=1}^{T}\sum_{k\in\mathcal{F}_t}\langle\nabla f_k(x_k), x_t - x^*\rangle - \sum_{t=1}^{T}\gamma B_\psi(x^*; x_t)$$
$$\leq \sum_{t=1}^{T}\frac{|\mathcal{F}_t|^2 G_\star^2}{2\sigma\gamma t} + \sum_{t=1}^{T}\left[(t-1)\gamma B_\psi(x^*; x_t) - t\gamma B_\psi(x^*; x_{t+1})\right]$$
$$+ \sum_{t=1}^{T}\left(|\mathcal{F}_t|G_\star + \frac{4\xi RG_\star}{\eta_t}\right)\|x_{t+1} - y_{t+1}^*\|$$
$$\leq \sum_{t=1}^{T}\frac{d|\mathcal{F}_t|G_\star^2}{2\sigma\gamma\sum_{\tau=1}^{t}|\mathcal{F}_\tau|} + \sum_{t=1}^{T}\left(|\mathcal{F}_t|G_\star + \frac{4\xi RG_\star}{\eta_t}\right)\|x_{t+1} - y_{t+1}^*\|$$
$$\leq \frac{dG_\star^2(1+\ln T)}{2\sigma\gamma} + \sum_{t=1}^{T}\left(|\mathcal{F}_t|G_\star + \frac{4\xi RG_\star}{\eta_t}\right)\|x_{t+1} - y_{t+1}^*\|. \tag{74}$$

Consider the approximate solution of $x_{t+1}$, we have

$$\sum_{k\in\mathcal{F}_t}\langle\nabla f_k(x_k), x_{t+1}\rangle + \frac{1}{\eta_t}B_\psi(x_{t+1}; x_t) \leq \sum_{k\in\mathcal{F}_t}\langle\nabla f_k(x_k), y_{t+1}^*\rangle + \frac{1}{\eta_t}B_\psi(y_{t+1}^*; x_t) + \rho_t.$$

We make $D_t(x) = \sum_{k\in\mathcal{F}_t}\langle\nabla f_k(x_k), x\rangle + \frac{1}{\eta_t}B_\psi(x; x_t)$. It is fact that $D_t$ is $\frac{\sigma}{\eta_t}$-strongly convex with respect to norm $\|\cdot\|$, then

$$D_t(x_{t+1}) - D_t(y_{t+1}^*) \geq \langle\nabla D_t(y_{t+1}^*), x_{t+1} - y_{t+1}^*\rangle + \frac{\sigma}{2\eta_t}\|x_{t+1} - y_{t+1}^*\|^2$$
$$\geq \frac{\sigma}{2\eta_t}\|x_{t+1} - y_{t+1}^*\|^2.$$

Combining above and $\rho_t = \frac{\eta_t^3}{2\sigma}$ gives

$$\|x_{t+1} - y_{t+1}^*\| \leq \sqrt{\frac{2\eta_t\rho_t}{\sigma}} = \frac{\eta_t^2}{\sigma}. \tag{75}$$

Substituting Eq. (75) into Eq. (74), we obtain the result of Lemma 12.

## G.2 Proof of Lemma 13

Due to the fact that $\psi(\cdot)$ is $\sigma$-strongly convex with respect to norm $\|\cdot\|$. Thus

$$B_\psi(y_{t+1}^*; y_t^*) + B_\psi(y_t^*; y_{t+1}^*) \geq \sigma\|y_t^* - y_{t+1}^*\|^2. \tag{76}$$

Meanwhile, based on the definition of Bregman divergence, the L.H.S of Eq. (76) can be rewritten as

$$B_\psi(y_{t+1}^*; y_t^*) + B_\psi(y_t^*; y_{t+1}^*) = \langle\nabla\psi(y_t^*) - \nabla\psi(y_{t+1}^*), y_t^* - y_{t+1}^*\rangle.$$

Recall back the optimality condition for the update rule, for any $y_t^*$ we have

$$\left\langle\eta_t\sum_{k\in\mathcal{F}_t}\nabla f_k(x_k) + \nabla\psi(y_{t+1}^*) - \nabla\psi(x_t), y_t^* - y_{t+1}^*\right\rangle \geq 0.$$

Rearranging the terms of the above equality yields

$$
\begin{aligned}
\left\langle \nabla\psi(\boldsymbol{y}_t^*) - \nabla\psi(\boldsymbol{y}_{t+1}^*), \boldsymbol{y}_t^* - \boldsymbol{y}_{t+1}^* \right\rangle \leq & \eta_t \sum_{k\in\mathcal{F}_t} \left\langle \nabla f_k(\boldsymbol{x}_k), \boldsymbol{y}_t^* - \boldsymbol{y}_{t+1}^* \right\rangle \\
& + \left\langle \nabla\psi(\boldsymbol{y}_t^*) - \nabla\psi(\boldsymbol{x}_t), \boldsymbol{y}_t^* - \boldsymbol{y}_{t+1}^* \right\rangle \\
\leq & \eta_t |\mathcal{F}_t| G_\star \|\boldsymbol{y}_t^* - \boldsymbol{y}_{t+1}^*\| \\
& + \xi G_\star \|\boldsymbol{x}_t - \boldsymbol{y}_t^*\| \cdot \|\boldsymbol{y}_t^* - \boldsymbol{y}_{t+1}^*\|.
\end{aligned}
\tag{77}
$$

Considering Eq. (76) and Eq. (77) gives

$$
\|\boldsymbol{y}_t^* - \boldsymbol{y}_{t+1}^*\| \leq \frac{\eta_t |\mathcal{F}_t| G_\star + \xi G_\star \|\boldsymbol{x}_t - \boldsymbol{y}_t^*\|}{\sigma}.
$$

Then combining with Eq. (75), we have the result of Lemma 13

$$
\begin{aligned}
\|\boldsymbol{x}_{t+1} - \boldsymbol{x}_t\| \leq & \|\boldsymbol{y}_{t+1}^* - \boldsymbol{y}_t^*\| + \|\boldsymbol{x}_{t+1} - \boldsymbol{y}_{t+1}^*\| + \|\boldsymbol{x}_t - \boldsymbol{y}_t^*\| \\
\leq & \frac{\eta_t |\mathcal{F}_t| G_\star + \eta_t^2 + \eta_{t-1}^2}{\sigma} + \frac{\eta_{t-1}^2 \xi G_\star}{\sigma^2}.
\end{aligned}
$$

