# OpenReview forum: "Online Sequential Decision-Making with Unknown Delays"
_ACM.org/TheWebConf/2024/Conference — TheWebConf24 Oral_

### Official Review · Reviewer_RRYi · 2023-10-31

**Novelty:** 4
**Technical Quality:** 4

**Review:**

This submission tries to study online sequential decision making of unknown delays, which is based on the traditional OCO methods to propose three types of online algorithms

pros
1. this draft is easy to follow
2. your core idea of follow the delayed regularized leader is reasonable
3. your online solutions is similar and closely linked with multi-armed bandits, like clustering based on exploration and exploitation, etc

cons
1. a major concern is that your current draft has no impressive empirical results to support
2. missing strong baselines, some related state-of-the-art sequential decision making methods are not tested
3. what's the practical usefulness of this proposed method is unclear

Overall, it's enjoyable to have a read on this draft, it would be glad to recommend towards acceptance, after this work thoroughly be polished based on all comments.

**Questions:**

1. your real-world data source is missing, try to add at least one or two
2. the number of data points or scale in your experiments needs to be verified
3. try to introduce some production data from companies, etc, to better demonstrate the usability of the proposed approach
4. there are related state-of-the-art you may want to compare: Distributed Clustering of Linear Bandits in Peer to Peer Networks, Fast Distributed Bandits for Online Recommendation Systems

**Reviewer Confidence:**

4: The reviewer is certain that the evaluation is correct and very familiar with the relevant literature

**Scope:**

4: The work is relevant to the Web and to the track, and is of broad interest to the community

---

### Official Review · Reviewer_NfNb · 2023-11-27

**Novelty:** 4
**Technical Quality:** 5

**Review:**

The paper studies the problem of online sequential decision making with unknown delays. The paper proposes three types of online algorithms to handle delayed feedback. The first one is for the full information feedback of loss functions. The contribution here is to replace the strong convexity with a more general notion of relative strong convexity. The second one is  the gradient information feedback where the proposed methods work for various norms as opposed to just euclidean norms. In the third scenario, the feedback is limited to the value information of the loss functions' gradient. The contribution here is to show matching regret bounds compared to the full information case.

S1: The various delayed OCO settings considered in this paper are natural and relevant.

S2: The theoretical contributions for different settings are non-trivial and interesting.

S3: The paper is generally well-written.

W1: The real world examples for various types of settings are missing. For example, in the full information feedback of loss function, what are the real-world examples where relative strong convexity is more suitable than strong convexity?

W2: For the gradient information feedback, the proposed methods work with various universal norms. What are some example use-cases where non-euclidean norms are used?

W3: The authors claim that the theoretical analysis method is entirely different from the previous works. It would be beneficial to summarize the key technical contributions in the introduction.

W4: More generally speaking, in the absence of experimental evaluations, I feel more real-world examples are needed to establish the importance of the general settings studied in this paper.

**Questions:**

See W1-W4.

**Reviewer Confidence:**

2: The reviewer is willing to defend the evaluation, but it is likely that the reviewer did not understand parts of the paper

**Scope:**

3: The work is somewhat relevant to the Web and to the track, and is of narrow interest to a sub-community

---

### Official Review · Reviewer_zaGs · 2023-11-29

**Novelty:** 5
**Technical Quality:** 5

**Review:**

The paper focuses on online sequential decision-making problems, where there is an unknown delay in obtaining the feedback after a decision is made. The authors take three existing algorithms from the literature, namely Follow The Regularized Leader, Mirror Descent, and Simplified Mirror Descent, and propose learning algorithms that account for delay in obtaining the feedback. The authors handle delayed full function information, full gradient information, and value information of gradient at the decision point, respectively, and provide regret bounds. Moreover, their algorithms only require an approximate solution for the optimization step.

Pros:
1. Application driven problem setting which arises in real-life scenarios.
2. Good theoretical contributions.

Cons:
1. Although it is not necessary to have experiments in all the papers, but it would have been great if this paper had real-world experiments motivated from real-life applications.
2. The computational complexity of the algorithms is missing. Along with missing experiments, it becomes difficult to judge the utility of the paper.
3. Presentation can be improved (details below).

**Questions:**

I understood large parts of the paper, and I believe the paper contains some novel contributions. However, I have some questions on which the authors can rebut on.

1. In the introduction, the authors mention lines 215-216. These statements seem more of technical detail that can be covered later. Can the authors provide a real-life application/example to make the reader understand when and why things would break due to the use of gradients, and especially, Euclidean norm. Similar is the case for lines 222 onwards. Those are technical reasons. It would be helpful to see application-based examples in the introduction.

2. Is assumption 4 a definition? That is, the wording seems more like a definition to me; however, it is intended to say that f_t is a relative strongly convex with respect to \psi.

3. In formal notations, the authors present definitions and assumptions one after the other without mentioning the intuitive meaning and what they will be required for. It’s not a negative since the definitions provided are standard in literature; however, I wonder if a better way to present is explaining when they are required.

4.  Time complexity of the algorithms is not mentioned, which is important here. The authors claim that their algorithms require only approximate solution. It would be good to know the time complexity of exact solution and the approximate ones.

5. Sadly, there are no experiments in the paper. The contributions are theoretical. The paper lacks motivating applications as well as experiments. Although experiments are not "must to have" for such papers, but since the proposed algorithms contain approximate solutions, it makes the reader's life easier to grasp the contributions when experiments are present. Moreover, the research community may start using such algorithms once they see an application.


Minor:
1. Typo in line 158: Internet -> internet
2. Typo in line 271: we -> We

**Reviewer Confidence:**

4: The reviewer is certain that the evaluation is correct and very familiar with the relevant literature

**Scope:**

3: The work is somewhat relevant to the Web and to the track, and is of narrow interest to a sub-community

---

### Official Review · Reviewer_kTgD · 2023-11-29

**Novelty:** 6
**Technical Quality:** 7

**Review:**

The paper addresses the problem of delays in online sequential decision-making within the framework of online convex optimization (OCO). The key challenge considered is the unknown delay in receiving feedback after making a decision, a scenario common in practical applications such as portfolio management where returns may not be immediately available.
The paper introduces three families of delayed algorithms tailored to handle different types of received feedback. These algorithms are designed to be versatile and applicable to universal norms.

Strengths:
The theoretical contributions are further supported by concrete examples demonstrating the efficiency of each algorithm under different norms.
Eemphasizes the generality of the proposed algorithms by providing regret bounds under cases of general convexity and relative strong convexity for each type of algorithm.

Weaknesses:
The paper, while theoretically sound, lacks studies on applicability and real world case studies, and especially quantitative benefits of the proposed approach over others.

**Questions:**

1. How do the proposed algorithms compare with existing approaches like Delayed Online Gradient Descent (DOGD) and Delayed Online Gradient Descent for Strongly Convex functions (DOGD-SC)? Can you provide a more detailed comparison in terms of performance and applicability?

2. Could you provide a more intuitive explanation of the theoretical results and regret bounds? How do these results translate into practical implications, and can you elaborate on the significance of these findings?

**Reviewer Confidence:**

2: The reviewer is willing to defend the evaluation, but it is likely that the reviewer did not understand parts of the paper

**Scope:**

3: The work is somewhat relevant to the Web and to the track, and is of narrow interest to a sub-community

---

### Decision · Program_Chairs · 2024-01-22

**Decision:**

Accept (Oral)

**Comment:**

Overview
 The paper presents novel approaches within the online convex optimization (OCO) framework for addressing delays in online sequential decision-making. The authors introduce three families of delayed algorithms to handle different types of feedback, each adaptable to various norms and providing corresponding regret bounds. Theoretical underpinnings are the core of the paper, with a focus on general convexity and relative strong convexity. Following are the strengths and weaknesses proposed by the reviewers.

 ## Common Strengths:
 1. Theoretical Contributions: All reviewers acknowledge the strong theoretical foundation of the paper, especially in providing regret bounds for different types of feedback and convexity conditions.
 2. Clarity and Structure: The paper is commended for its clarity and ease of comprehension.

 ## Concerns and Recommendations:
 1. Lack of Empirical Validation: A major concern shared by reviewers is the absence of empirical or real-world case studies to validate the theoretical findings. Reviewers also suggest providing a detailed comparison with existing approaches like DOGD and DOGD-SC, to understand the paper's contributions more concretely.
 2. Real-World Examples and Use-Cases: Reviewers express the need for real-world examples to contextualize the algorithms, especially in scenarios involving non-Euclidean norms and relative strong convexity.
 3. Computational Complexity Analysis: The computational complexity of the proposed algorithms is not discussed, which is crucial for assessing their practical utility.
 4. Clarification of Technical Contributions: The paper should more clearly outline its key technical contributions, particularly in comparison to previous works.

 Based on my reading, the authors' response has addressed the concerns 3 and 4, and partly 1 by introducing some empirical validation. One real-world dataset is used yet limited. Given this, the authors are suggested to provide more serious real-world examples and evaluation to motivate the proposed methods.